# Biotechnologically produced chitosans with nonrandom acetylation patterns differ from conventional chitosans in properties and activities

Sruthi Sreekumar[1,2,3,5], Jasper Wattjes[1,2,5], Anna Niehues ®[1,5], Tamara Mengoni[1], Ana C. Mendes ®[2], Edwin R. Morris[4,6], Francisco M. Goycoolea ®[1,3] & Bruno M. Moerschbacher ®[1] ✉

Chitosans are versatile biopolymers with multiple biological activities and potential applications. They are linear copolymers of glucosamine and $N$-acetylglucosamine defined by their degree of polymerisation (DP), fraction of acetylation ($F_A$), and pattern of acetylation (PA). Technical chitosans produced chemically from chitin possess defined DP and $F_A$ but random PA, while enzymatically produced natural chitosans probably have non-random PA. This natural process has not been replicated using biotechnology because chitin de-$N$-acetylases do not efficiently deacetylate crystalline chitin. Here, we show that such enzymes can partially $N$-acetylate fully deacetylated chitosan in the presence of excess acetate, yielding chitosans with $F_A$ up to 0.7 and an enzyme-dependent non-random PA. The biotech chitosans differ from technical chitosans both in terms of physicochemical and nanoscale solution properties and biological activities. As with synthetic block co-polymers, controlling the distribution of building blocks within the biopolymer chain will open a new dimension of chitosan research and exploitation.

Chitosans are binary copolymers of partly hydrophobic $N$-acetylglucosamine (GlcNAc) and, at ca. pH 6 or below, cationic glucosamine (GlcN) linked by β−1,4-glycosidic bonds. In nature, chitosans are prominently found in the specialised fungal cell walls formed during host tissue penetration[1,2]. In the laboratory, chitosans can form gels, films, sponges, and nanoparticles and have been shown to interact with partly polyanionic/partly hydrophobic biomolecules such as proteins, nucleic acids, mucins, and phospholipid membranes[3,4]. Both the physicochemical properties and biological activities of chitosans are directly influenced by their degree of polymerisation (DP), molar fraction of acetylation ($F_A$), and, possibly, pattern of acetylation (PA)[5].

It is possible to synthesise chitosan with tight control of the DP and $F_A$ by homogeneous or heterogeneous chemical de-$N$-acetylation of chitin or $N$-acetylation of fully deacetylated chitosan, i.e., 1→)-β-D-Glc$p$N-(→4 (in the following referred to as polyglucosamine for brevity), so that the influence of these parameters can be studied in detail[6–8]. The structure-function relationships thus derived, summarised in the chitosan matrix which visualises bioactivities as a function of DP and $F_A$[5], have allowed the development of chitosan-based applications for food preservation and plant protection, exploiting the antimicrobial properties of chitosans and their ability to induce plant defence responses[9,10]. However, biomedically relevant

[1]Institute for Biology and Biotechnology of Plants, University of Münster, 48143 Münster, Germany. [2]Research Group for Food Production Engineering, Laboratory of Nano-BioScience, National Food Institute, Technical University of Denmark, 2800 Kgs Lyngby, Denmark. [3]School of Food Science and Nutrition, University of Leeds, LS2 9JT Leeds, United Kingdom. [4]School of Food and Nutritional Sciences, University College Cork, Cork, Ireland. [5]These authors contributed equally: Sruthi Sreekumar, Jasper Wattjes, Anna Niehues. [6]Deceased: Edwin R. Morris. ✉e-mail: moersch@uni-muenster.de

activities of chitosans still remain poorly understood on a molecular or nanoscale level, so that the development of chitosan-based biomaterials for medical applications lags behind the presumed potential of these functional biopolymers, possibly because of the unknown role of PA in the interaction of chitosans with human or animal cells[5].

In contrast to DP and $F_A$, the PA is more difficult to analyse and control[11]. Initially, chitosans produced by the heterogeneous de-$N$-acetylation of chitin were thought to possess blocks of acetyl groups, whereas chitosans produced by homogeneous de-acetylation or $N$-acetylation were predicted to have a more random PA[12-14]. However, [13]C nuclear magnetic resonance (NMR) spectroscopy showed that dyad frequencies of commercially available chitosans do not differ from what is expected for random PA, regardless of the production method[15,16]. It has therefore been impossible to investigate the potential influence of acetylation patterns on the properties and activities of chitosan polymers, but theoretical considerations strongly suggest that such an influence exists[5]. Importantly, natural chitosans are likely to possess non-random acetylation patterns due to their enzymatic route of synthesis.

One way in which PA could influence the biological activity of chitosans is via the sequence-dependent activity of chitosanolytic enzymes[17,18]. Partially acetylated chitosans can be depolymerised by both chitinases and chitosanases. The former are ubiquitous whereas the latter appear to be restricted to microorganisms. Both types of enzyme tend to have more or less pronounced sequence specificity because their substrate-binding cleft consists of multiple subsites, each binding a single monomeric subunit and showing a preference for either GlcN or GlcNAc[19-21]. The PA of the substrate therefore determines its position in the binding site, in turn specifying GlcN or GlcNAc units at and near the reducing and non-reducing ends of the oligomeric products. The PA of the substrate and the subsite specificities and preferences of the enzyme therefore combine to determine how readily the polymer is degraded and the types of oligomers produced. This is important because the biological activity of chitosan polymers in a given tissue may be exerted by the polymer itself, such as by electrostatic interactions with the plasma membrane of a target cell[6], or by enzymatic degradation products generated in situ, which may be recognised by specific receptors[8,22-25]. In both cases, the PA can be expected to strongly influence the pharmacodynamics and pharmacokinetics of the chitosan polymers, such as their bioavailability, their metabolic fate, their absorption, distribution, and excretion from a target tissue and organism, as well as their local or systemic, immediate or retarded, long- or short-lived action in humans, animals, or plants[5].

Although chemically produced conventional chitosans feature a random PA, natural chitosans produced by chitin deacetylases (CDAs) may possess non-random acetylation patterns[26-28]. However, CDAs are almost inactive on native chitin polymers in vitro probably because the substrate is crystalline[29,30]. In contrast, CDAs do act on soluble chitosan polymers with a high $F_A$, converting them to low-$F_A$ chitosans[30-32]. Such chemo-enzymatically produced chitosans can exhibit non-random acetylation patterns[27,28]. Depending on which CDA is used, the PA ranges from Bernoullian randomness to a block-like or more regular distribution of acetyl groups, as confirmed by [13]C-NMR dyad analysis and enzymatic mass spectrometry (EMS) fingerprinting[27]. However, the $P_\Sigma$ values did not differ greatly from 1, the value expected for random PA[15], because the high-$F_A$ chitosan polymer used as a substrate was produced chemically and thus had a random PA. Accordingly, the products also tended towards randomness, with $P_\Sigma$-values of 0.75–1.31[27]. To address this issue, we approached the problem from the opposite perspective, using CDAs in reverse to partially $N$-acetylate polyglucosamine in the presence of excess acetate as a means to retain their regioselectivity and generate chitosans with a specific PA, as shown previously for GlcN oligomers[17,33,34].

Here, we show that different recombinant chitin deacetylases are able to $N$-acetylate polyglucosamine, yielding partially acetylated chitosans ranging in $F_A$ all the way up to 0.7, i.e. in the full range of acid-soluble chitosans. Depending on the enzyme used, the acetylation patterns of these biotech chitosans can differ from randomness towards a more block-wise or a more regular distribution. Extensive functional tests show that these biotech chitosans possess different physicochemical properties and biological activities compared to conventional technical chitosans with random acetylation.

## Results

### CDA can $N$-acetylate polyglucosamine, yielding high-$F_A$ chitosans with non-random acetylation

We tested four recombinant fungal CDAs to determine whether they would act in reverse on polyglucosamine, producing partially acetylated chitosan polymers in the presence of excess acetate (Fig. 1). This yielded chitosans with a $F_A$ as high as 0.7 (Supplementary Fig. 1a) and the process could be controlled by adjusting the reaction conditions such as the incubation time or enzyme concentration, giving us access to the full spectrum of soluble chitosan polymers (Fig. 1a). EMS fingerprinting revealed that the enzymatically and chemically $N$-acetylated chitosans differed in terms of PA, with the former showing average GlcN and GlcNAc block sizes that deviated from random PA chitosans in different directions depending on which enzyme we used. Detailed analysis of the chitinosanase products clearly showed that chitosans with a similar $F_A$ produced by different enzymes showed significantly different properties, confirming that each enzyme generated chitosan polymers with a unique PA (Fig. 1b). The oligomeric products of PgtCDA most closely resembled those of the conventional random-PA chitosan, whereas the products of AnCDA and PesCDA resembled each other but clearly differed from those of the conventional chitosan, featuring larger oligomers that indicated block-wise rather than random acetylation. In contrast, the products of CnCDA4 were dominated by smaller oligomers, indicating regular rather than random acetylation. A comparative analysis of block size distributions revealed that each enzyme generated unique products that differed from the chemical control (Supplementary Fig. 1b). Interestingly, these differences were already visible at early time points in the $N$-acetylation reaction when the $F_A$ was low. We scaled up the PesCDA reaction to yield sufficient product for [13]C-NMR dyad analysis. We initially used a sample with a $F_A$ of 0.5 but were unable to dissolve it in $D_2O$ following treatment with $HNO_2$. Unlike chemically $N$-acetylated chitosan polymers with the same $F_A$[35], the enzymatically $N$-acetylated chitosan polymer could not be solubilised using a stoichiometric quantity of acid but required 5% acetic acid, again indicating a non-random PA. This issue was not encountered with an enzymatically (also using PesCDA) $N$-acetylated chitosan with a $F_A$ of 0.33, for which [13]C-NMR dyad analysis revealed a $P_\Sigma$ value of 0.3, verifying the strong block-wise distribution of acetyl groups (Fig. 1c; Supplementary Fig. 1c).

All chitosans used in this study are named as CS.xx[Y-Ac], where.xx represents the $F_A$ of the chitosan and the superscript denotes the type of $F_A$-modification applied, with D = chemical de-acetylation, N = chemical $N$-acetylation, and E = enzymatic $N$-acetylation.

### PA influences the physicochemical properties of chitosans in solution

Next, we compared the physicochemical solution properties of the enzymatically $N$-acetylated $P_\Sigma = 0.3$ block-PA chitosan polymer ($F_A = 0.33$, DP = 800, dispersity (Đ) = 1.9) with those of a chemically $N$-acetylated $P_\Sigma = 1.0$ random-PA chitosan ($F_A = 0.34$, DP = 700, Đ = 1.8) (Fig. 2). The intrinsic viscosity of the biotech chitosan in water was significantly lower than that of the conventional chitosan (Supplementary Table 1). This difference was strongly reduced in the presence of 0.1 M NaCl, indicating a more densely packed conformation in

solution presumably caused by hydrophobic interactions between poly-GlcNAc blocks. This solution behaviour was similar to that reported for a chemo-enzymatically prepared chitosan polymer with slightly more block-wise than random PA ($P_\Sigma = 0.75$)[27]. While the conventional chitosan readily formed nanoparticles by ionic gelation with tripolyphosphate (TPP)[36,37], particle formation was limited with the biotech chitosan (Fig. 2a). The Z-average hydrodynamic diameter of the random-PA chitosan nanoparticles increased with the chitosan:TPP

ratio, whereas that of the scarce block-PA chitosan nanoparticles remained surprisingly stable, again possibly reflecting hydrophobic interactions between poly-GlcNAc blocks which are not influenced by the molar charge ratio. Block-PA nanoparticles showed a higher dispersity in size than random-PA nanoparticles, often exhibiting a bimodal distribution indicative of the formation of small aggregates (Supplementary Fig. 2). For both types of chitosan, the surface charge increased with increasing $NH_2$/TPP ratio from 1.2 to 2.1, reaching ca.

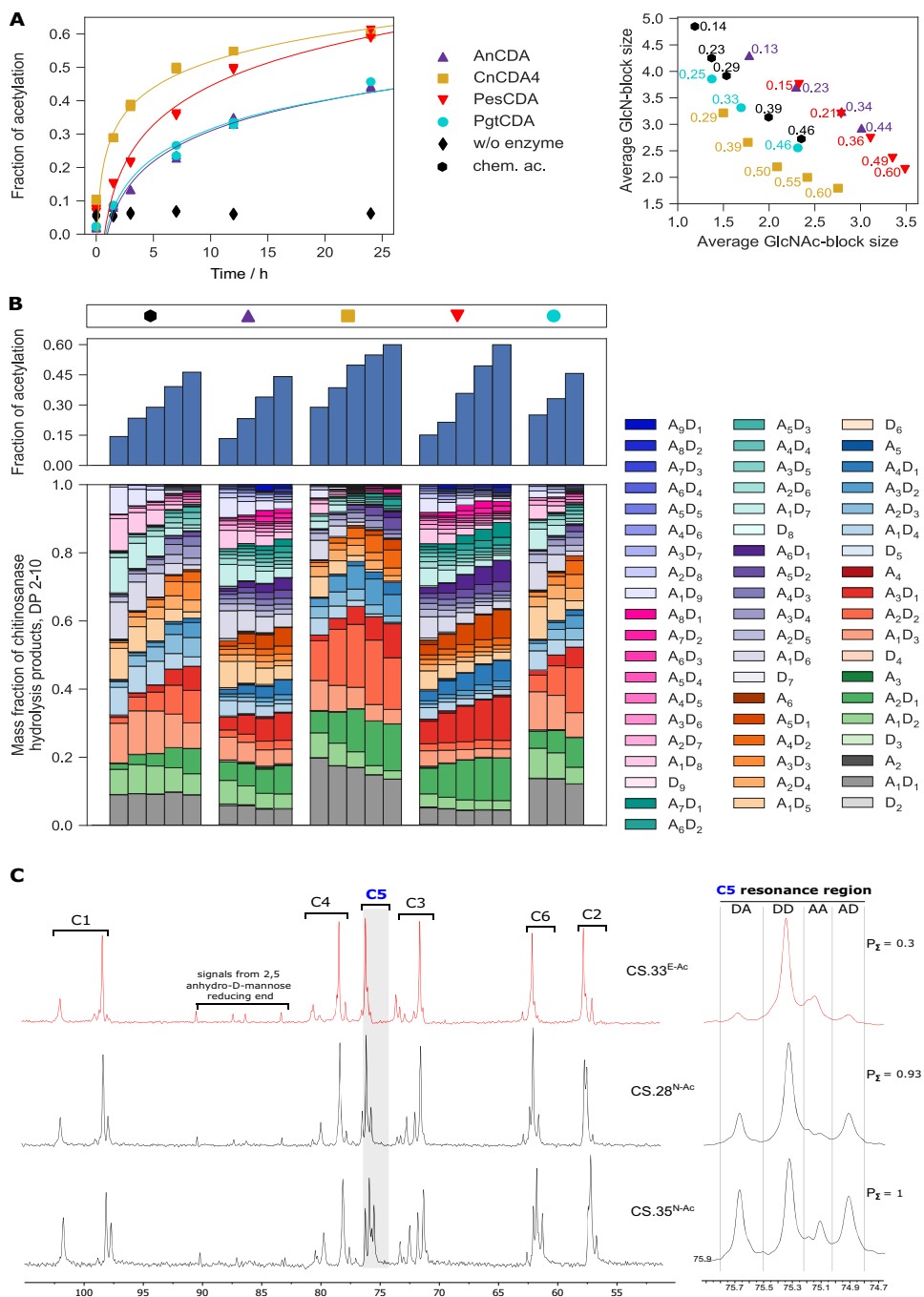

**Fig. 1 | CDAs can *N*-acetylate polyglucosamine, yielding high-$F_A$ chitosans with non-random PA. A** Polyglucosamine ($F_A = 0.03$) was incubated in the presence of 1.5 M sodium acetate for 24 h (pH 7.5) with four different recombinant fungal CDAs (AnCDA from *Aspergillus niger*, CnCDA4 from *Cryptococcus neoformans*, PgtCDA from *Puccinia graminis* f. sp. *tritici*, and PesCDA from *Pestalotiopsis* sp.) or without enzyme as a control. The $F_A$ (left panel) and average block sizes of DP 2–10 (right panel) in the resulting chitosan polymers were analysed using chitinosanase-based EMS fingerprinting. Chemically *N*-acetylated chitosans were used as positive controls. **B** The chemically and enzymatically *N*-acetylated chitosans (see **A**) were hydrolysed with chitinosanase, and products of DP 2–10 were analysed by HILIC-ESI-MS to calculate the $F_A$ of the polymeric substrates (upper panel) and characterise the oligomeric products (lower panel). **C** $^{13}$C-NMR analysis of one enzymatically (CS.33$^{E-Ac}$) produced (using PesCDA) and two chemically *N*-acetylated chitosans with a similar $F_A$ (CS.28$^{N-Ac}$ and CS.35$^{N-Ac}$) (left panel) and magnification of the C-5 resonance region (right panel). The four dyad peak areas ($I_{DA}$, $I_{DD}$, $I_{AA}$, and $I_{AD}$) were integrated and used to calculate $P_\Sigma$ values[15].

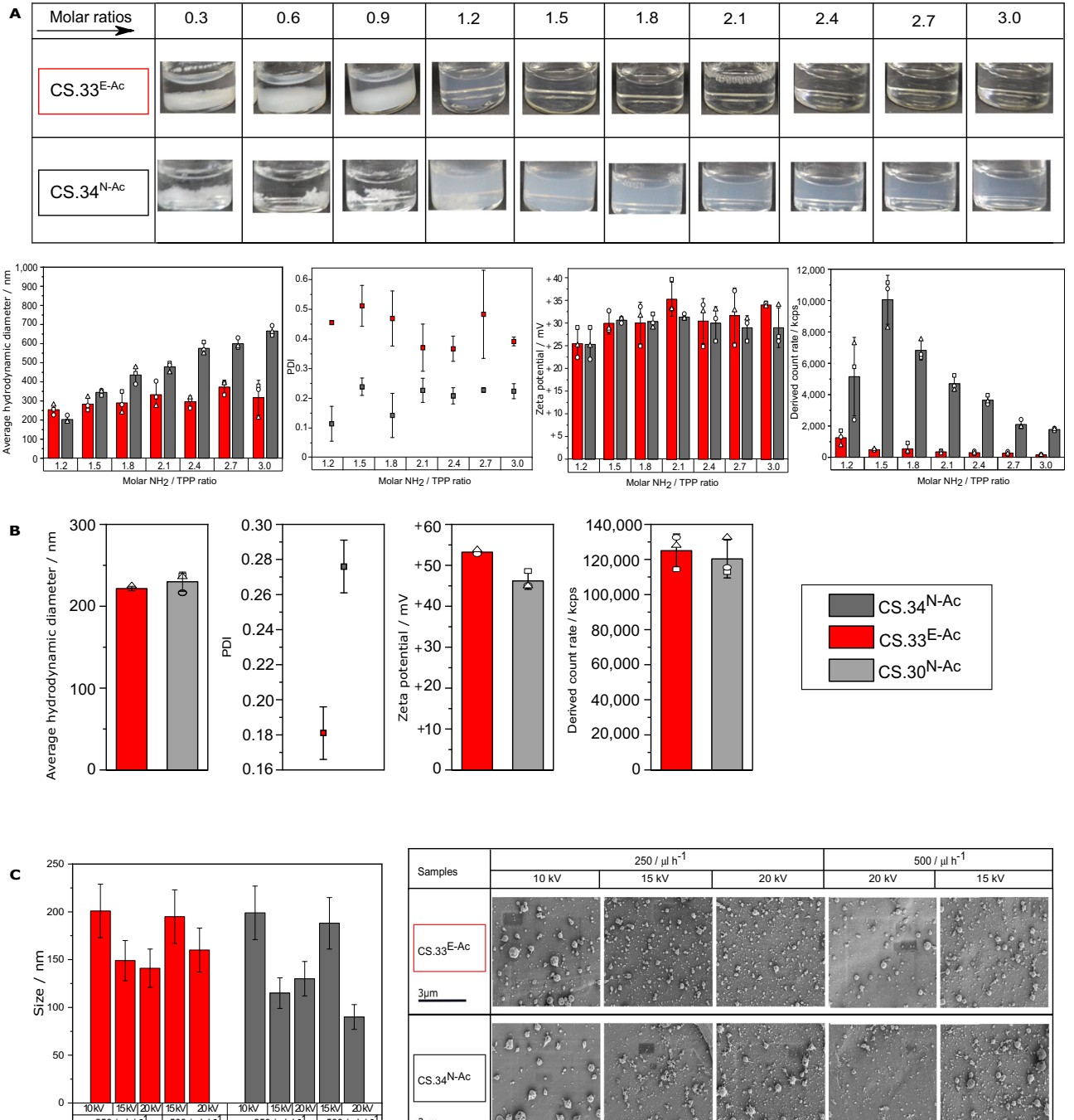

**Fig. 2 | PA influences the nanoformulation of chitosans. A** Physicochemical characterisation of chitosan-TPP nanoparticles prepared by ionic gelation at different NH2/TPP molar ratios: images of chitosan-TPP formulations after preparation (upper panel), Z-average hydrodynamic diameter and polydispersity index (PDI) (lower left panel), zeta potential (lower centre panel), and derived count rate (lower right panel) determined by dynamic light scattering. Data represent three independent experiments plotted as means ± SD. **B** Physicochemical characterisation of chitosan nanocapsules prepared by solvent displacement: average hydrodynamic diameter and PDI (left panel), zeta potential (centre panel), and derived count rate (right panel) determined by dynamic light scattering. Data represent three independent experiments plotted as means ± SD. **C** Characterisation of electrosprayed chitosan particles: size of particles determined from scanning electron micrographs of 50 particles each (right panel) using ImageJ software, given as means ± SE (left panel). Two preceding, preliminary visualisations using a benchtop-SEM gave similar results, but were not quantified.

+30 mV and remaining constant at higher ratios, as expected. In contrast, both types of chitosan readily yielded nanocapsules (chitosan-coated nanoemulsions[38]) with no significant differences between the two forms (Fig. 2b). Whereas chitosan forms the bulk of chitosan-TPP nanoparticles, the nanocapsules feature a thin chitosan surface layer, apparently with less influence on the overall properties of the system. We also found that both types of chitosan were able to form nanoparticles prepared using an electrospray technique (Fig. 2c) in the presence of 30% acetic acid and 30% ethanol[39,40]. These conditions are likely to reduce the influence of electrostatic and hydrophobic interactions.

The quantitative solution properties of the two chitosan polymers (Fig. 3) were compared by measuring the small deformation rheology characterised by the frequency dependence profile of the elastic (G′)

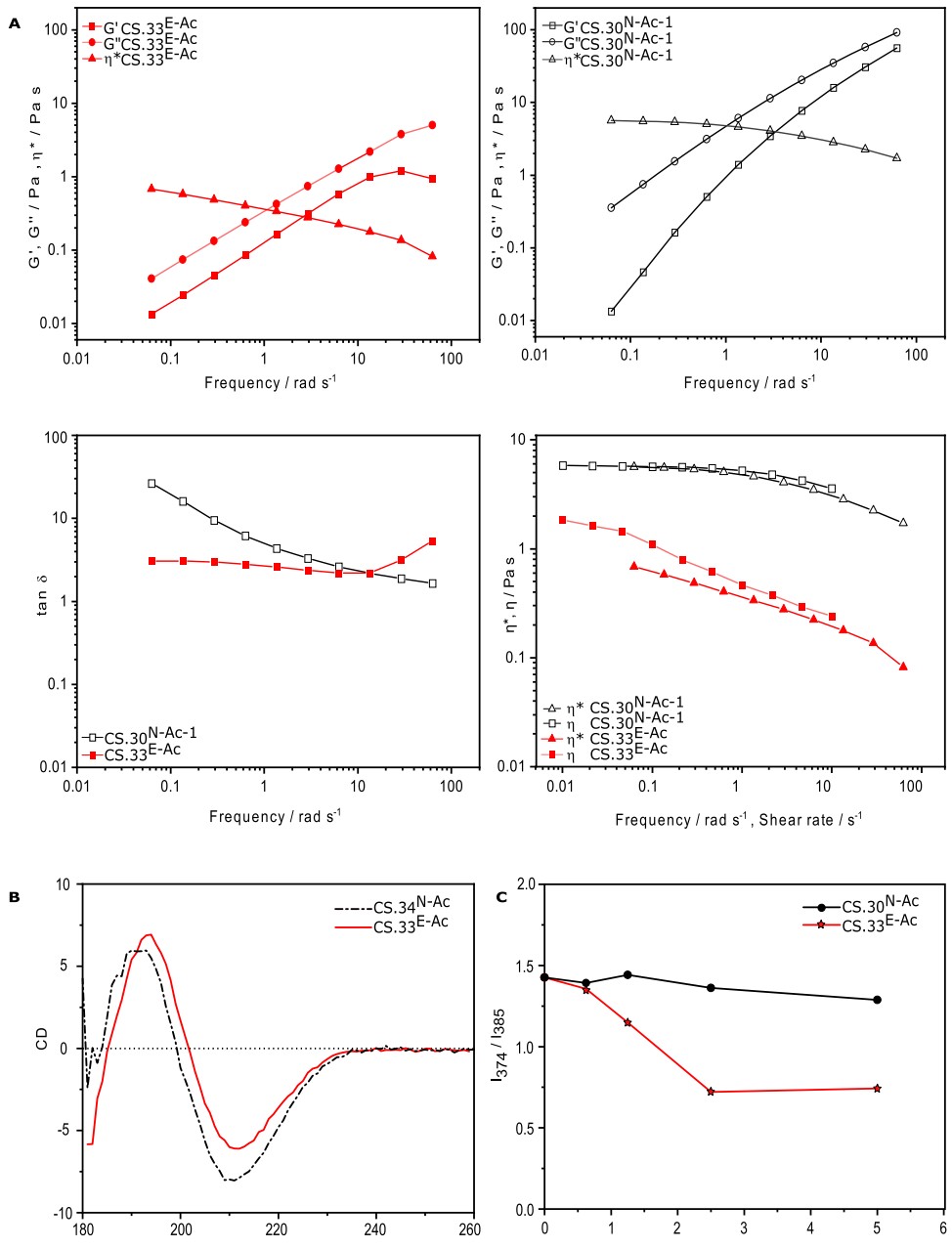

**Fig. 3 | PA influences the properties of chitosans in solution. A** Rheological analysis of enzymatically (CS.33$^{E-Ac}$) and chemically (CS.30$^{N-Ac-1}$) produced chitosan polymers prepared from 30 mg ml$^{-1}$ chitosan dissolved in a 5% stoichiometric excess of acetic acid at 25 °C: dependence of viscoelastic moduli G′ and G″ and complex viscosity ($\eta$*) on the frequency (strain = 20%) of CS.30$^{E-Ac}$ (upper left panel) and CS.30$^{N-Ac-1}$ (upper right panel); dependence of tan δ (=G″/G′) on frequency (strain = 20%) (lower left panel) and Cox-Merz representation of $\eta$* and steady-shear viscosity ($\eta$) as a function of frequency and shear rate, respectively (lower right panel). All measurements were conducted within the linear viscoelastic region. **B** Circular dichroism spectroscopy of enzymatically (CS.33$^{E-Ac}$) and chemically (CS.34$^{N-Ac}$) produced chitosan (0.5 mg ml$^{-1}$ at 25 °C). **C** Ratio of pyrene fluorescence emission intensities ($I_{374}/I_{385}$) from enzymatically (CS.33$^{E-Ac}$) and chemically (CS.30$^{N-Ac}$) produced chitosan as a function of chitosan concentration ($\lambda_{ex}$ = 343 nm, 2 μM pyrene, 5% stoichiometric excess of acetic acid, 100 mM NaCl, 25 °C).

and loss (G″) moduli and the complex viscosity ($\eta$*). Intriguingly, the profile of the conventional random-PA chitosan was typical of random coil behaviour in a dilute polymer solution[41], whereas the biotech block-PA chitosan showed overall lower G′ and G″ values but the frequency dependence of both moduli displayed a similar slope (Fig. 3a). This peculiarity was also illustrated by the frequency dependence of tan δ (=G″/G′) for both solutions, which showed a monotonic decrease for the conventional chitosan, but was essentially independent of frequency for the biotech chitosan apart from an abrupt upturn above ca. 10 rad s$^{-1}$, which corresponds to an anomalous downturn in G′ and

probably arises from the onset of resonance in the measuring geometry of the rheometer at high frequency. Figure 3a also shows the corresponding Cox-Merz superposition for both systems, representing the values of $\eta$* and the steady-shear viscosity ($\eta$) as a function of oscillation frequency and shear rate, respectively. Notably, the random-PA polymer showed almost Newtonian behaviour (with $\eta$* and $\eta$ showing little dependence on frequency and shear rate, respectively) and similar values of both viscosities, thus conforming to the Cox-Merz rule[42]. In contrast, the values of $\eta$* and $\eta$ were almost an order of magnitude lower for the block-PA polymer and yet it displayed much

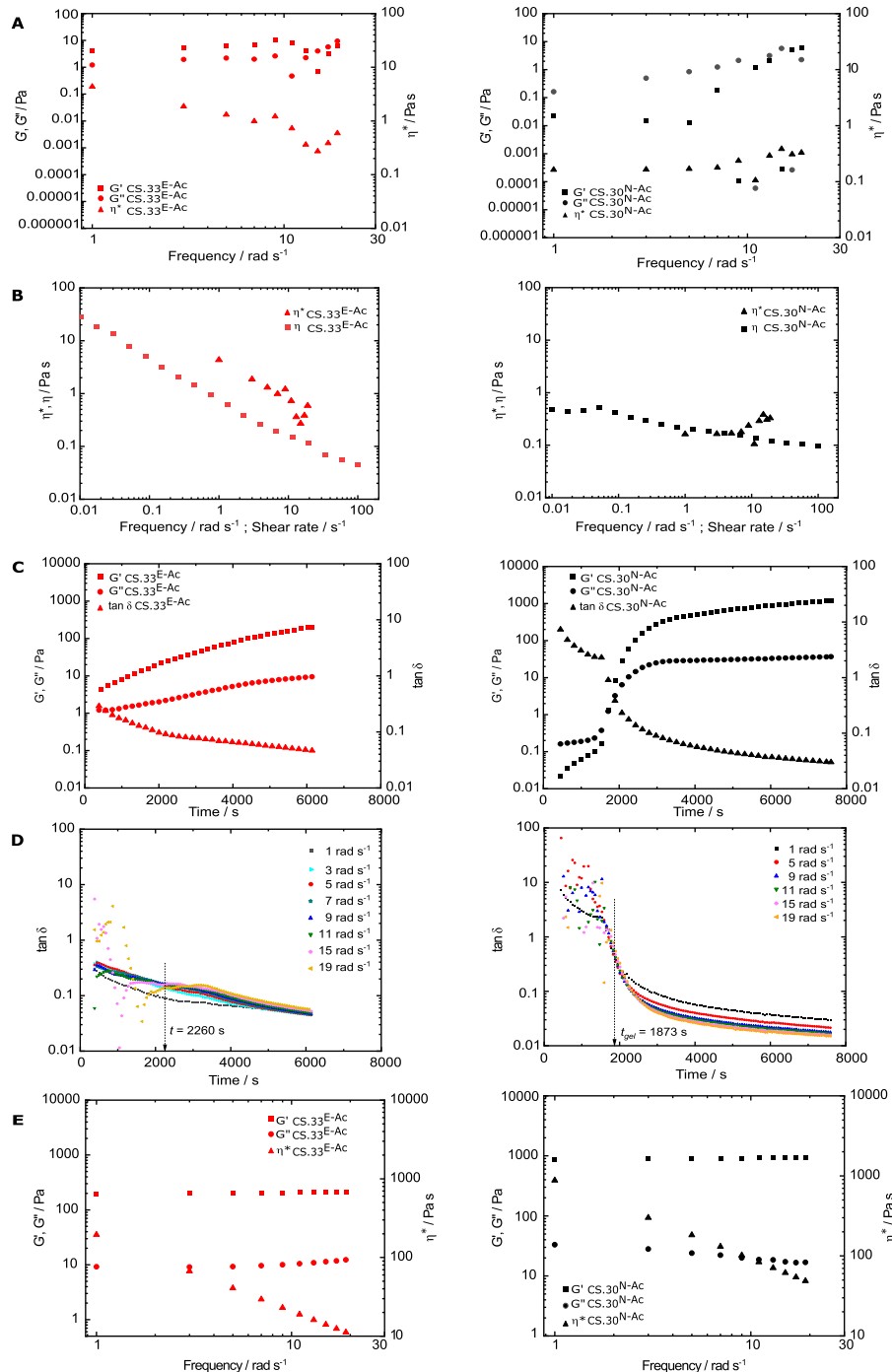

**Fig. 4 | PA influences the gelation properties of chitosans. A** Dependence of the viscoelastic moduli G′ and G″ and of the complex viscosity ($\eta^*$) on frequency for CS.33$^{E-Ac}$ (left panel) and CS.30$^{N-Ac}$ (right panel) (strain 5%, 40 °C) in 0.5 M acetate buffer pH 4.5 (ca. 14 mg mL$^{-1}$ chitosan) crosslinked with genipin (genipin/GlcN molar ratio = 0.5). **B** Cox-Merz superpositions of steady-shear viscosity ($\eta$) and complex viscosity ($\eta^*$) for CS.33$^{E-Ac}$ (left panel) and CS.30$^{N-Ac}$ (right panel). **C** Dependence of the viscoelastic moduli G′ and G″ and of the liquid-like character (tan δ) on time for CS.33$^{E-Ac}$ (left panel) and CS.30$^{N-Ac}$ (right panel) (strain 5%, 40 °C)

in 0.5 M acetate buffer pH 4.5 (ca. 14 mg mL$^{-1}$ chitosan) crosslinked with genipin (genipin/GlcN molar ratio = 0.5). **D** Multiwave variation of tan δ(t) at the fundamental and multiwave harmonic frequencies (as shown in labels), showing the closest earliest crossover time for CS.33$^{E-Ac}$ (left panel) and the critical gel time ($t_{gel}$) for CS.30$^{N-Ac}$ (right panel). **E** Endpoint (time 6000 s) dependence of the viscoelastic moduli G′ and G″ and of the complex viscosity ($\eta^*$) on frequency for CS.33$^{E-Ac}$ (left panel) and CS.30$^{N-Ac}$ (right panel).

greater shear thinning behaviour, while still conforming to the Cox-Merz rule within experimental error. These results suggest that the conventional chitosan behaves, as expected, as a polymer random coil in the entangled regime, whereas the biotech chitosan adopts a different conformation. This difference in conformation was also indicated by circular dichroism (CD) spectroscopy, which revealed subtle

but important differences in the peak bands of the two samples (Fig. 3b). Indeed, both the peak and trough CD bands of the block-PA chitosan appeared shifted by ca. 4 nm to a higher λ than the random-PA chitosan. Finally, we investigated the dependence of pyrene fluorescence ($I_{374}/I_{385}$ ratio) on the concentration of both chitosan solutions[43,44] (Fig. 3c). The block-PA chitosan had a much stronger

dependency than the random-PA chitosan, thus confirming the greater hydrophobicity of the block-PA polymer. The results support our hypothesis that the biotech chitosan contains a greater proportion of more hydrophobic block domains than the conventional polymer.

The above results suggest that the gelling behaviour of the enzymatically produced, block-PA chitosan will differ from that of conventional, random-PA chitosans. Therefore, we next compared the process of gelation of conventional random-PA and biotech block-PA chitosans by covalent genipin crosslinking (Fig. 4). While the viscoelastic profile of the random-PA sample during the first minutes after genipin addition shows the typical behaviour of an entangled polymer solution as already seen in Fig. 3a, that of the block-PA sample differs sharply, with G′ predominating over G″ even at the lowest frequency, and both moduli showing an increasing dependence on frequency up to 10 rad s$^{-1}$, beyond which the appearance of geometry resonance leads to a more erratic behaviour (Fig. 4a). The different rheological behaviour of the two types of chitosan at the outset of the gelation experiment can also be seen in the Cox-Merz superpositions of $\eta$ and $\eta^*$ data (Fig. 4b). Interestingly, while the random-PA sample conforms very closely with the Cox-Merz rule and again agrees with the results shown in Fig. 3a, the block-PA sample violates this rule, with the values of $\eta^*$ lying on top of those of $\eta$, characteristic of a weak gel system such as concentrated aqueous xanthan gum solutions[45]. The rheological behaviour of the biotech block-PA chitosan was unexpected and suggests that the chitosan solution started to show a weak gel behaviour at the outset of the rheological measurements. The most likely explanation for this result is that the addition of the small aliquot of the ethanolic genipin solution drove the chitosan chains with their blocks of consecutive acetylated GlcNAc units to self-association, resulting in the formation of a tenuous gel network and immediately starting the covalent crosslinking.

The kinetics of gelation of the two chitosan samples is visualised by the evolution of G′, G″, and tan δ over time registered at the fundamental frequency (1 rad s$^{-1}$) (Fig. 4c). In the block-PA sample, G′ is greater than G″ from the outset of the experiment, and both moduli steadily and strongly increase, G′ eventually reaching ca. 200 Pa, while tan δ rapidly decreases from ca. 0.4 to ca. 0.04. Notably, an approximate visual interpolation of G′ and G″ traces to zero time reveals that G′ would continue to predominate over G″, thus suggesting that since the addition of the genipin aliquot to the chitosan solution, the system exhibited a weak gel behaviour. By contrast, the conventional random-PA chitosan shows a predominantly liquid-like behaviour at early times, with G″ predominating over G′ and a high tan δ value, followed by an abrupt sigmoidal increase of both moduli, and G′ surpassing G″ at ca. 1750 s, eventually reaching ca. 1000 Pa. This behaviour has all the hallmarks of a critical sol-to-gel transition as previously shown for genipin-crosslinked gelation of conventional chitosans[46,47]. To determine the critical gelation time ($t_{gel}$) with greater precision, we adopted the criterion of Winter and Chambon[48] and measured tan δ at varying frequencies using the multiwave harmonics programme. The registered tan δ traces showed a sudden drop from values greater than 1.0 to below 0.1, converging tightly into a single point at $t_{gel}$ of 1873 s (Fig. 4d). For the block-PA sample, it is impossible to pinpoint a single time point where all the tan δ(t) converge, and the trace at the fundamental frequency lies lower than those at all other frequencies throughout the entire period of the experiment. Again, this profile is in sharp contrast to the tan δ(t) traces registered for the random-PA sample.

Finally, we determined the endpoint mechanical spectra for both chitosan-genipin gels, which both showed the characteristic features of firmly set gels, namely a predominance of G′ over G″, with no dependence of either on frequency, and a negative dependence of $\eta^*$ with a slope of −1 (Fig. 4e). Endpoint amplitude (strain) sweeps on both gelled systems confirmed that the measurements were recorded well within the linear viscoelastic region assessed between 0.1 and 100%.

Both gels retained their mechanical strength up to a strain of 47.8%. Of note, however, the strain dependence of G′ and G″ moduli within this regime revealed subtle (though meaningful) differences between the two samples (Supplementary Fig. 3). G′ of the block-PA chitosan gel showed a monotonic increase in G′ from 233.6 to 395.5 Pa up to a strain of 46.8% ($\Delta = 69\%$) beyond which the gel started to yield at greater strains, while the corresponding increase in G″ was from 10.3 to 13.9 Pa ($\Delta = 35\%$) up to a strain of 14.9%, beyond which an abrupt increase up to 30.9 Pa ($\Delta = 122\%$) up to a strain of 68.7% was recorded. The increases in G′ and G″ for the random-PA chitosan gel, within the same strain range, were proportionally much smaller than those observed for the block-PA gel (1361.1 to 1622.4 Pa ($\Delta = 19\%$) and 38 to 44.9 Pa ($\Delta = 4\%$), respectively. The more pronounced increases in gel strength with increasing deformation observed in the block-PA sample with respect to the random-PA one is diagnostic for microstructural and possibly thermodynamic differences between both gel networks, as discussed below. The visual inspection of the vials containing the remaining solutions kept at 40 °C confirmed that despite the different time gelation profiles observed on both samples, both developed the characteristic blue colour of chitosan-genipin covalently crosslinked gels. After overnight storage of the gels kept in the glass vials under refrigeration, it was interesting that the block-PA gels suffered slightly greater synaeresis (i.e., expelling of water from the gel driven by its contraction) than the random-PA ones (Supplementary Fig. 4).

Clearly, while both random-PA and block-PA chitosans are undergoing the same chemical process that leads to the consolidation of a firmly set genipin-crosslinked gel network, the process follows very different kinetics in the two chitosans. Unlike the conventional random-PA chitosan, the biotech block-PA chitosan appears to start forming a gel immediately after the addition of genipin, possibly triggered by a self-association phenomenon upon the addition of ethanol (needed to dissolve genipin) driven by hydrophobic associations between the GlcNAc blocks and by the fast reaction step of the covalent genipin-chitosan crosslinking[49]. This weak network is subsequently consolidated into a firm gel network by covalent crosslinking with genipin, but eventually yielding a somewhat weaker gel than that formed more slowly by the random-PA chitosan. As the mechanical strength of a gel is governed by the number of elastically-active network chains (EANCs) per repeat unit in the gel network[50], such junctions apparently occur less frequently in the block-PA than in the random-PA gels. This could be diagnostic either of lower availability of amino groups able to participate in the crosslinking process in the former, as previously suggested for BSA-genipin compared with gelatine-genipin-crosslinked gels[49], or due to the formation of longer junctions created by the deacetylated GlcN blocks, resulting in a lower net number of EANCs than in the random-PA gels. Further studies using complementary techniques such as UV-vis spectroscopy and SAXS, and a range of different polymer and genipin concentrations, temperature, and block-PA samples of varying $F_A$ will contribute to elucidate the mechanisms at play.

## PA influences the biological activity of chitosans

We next compared the biological activities of the two chitosan samples (Fig. 5). The enzymatically N-acetylated block-PA chitosan showed stronger antibacterial activity than the chemically N-acetylated random-PA chitosan against the Gram-negative bacterium *Pseudomonas syringae* in liquid culture, while both chitosans were equally active against the biofilm-forming Gram-positive bacterium *Bacillus licheniformis* (Fig. 5a; Supplementary Fig. 5). The antibacterial activity of conventional chitosans against Gram-negative bacteria is inversely related to the $F_A$[10]. This probably reflects the decreasing charge density, because the antimicrobial activity of chitosans is thought to be caused by electrostatic interactions involving GlcN-rich blocks in the polymer chain that are large enough to disrupt membrane integrity[51,52]. The block-PA chitosan appears to contain sufficiently large GlcN-rich

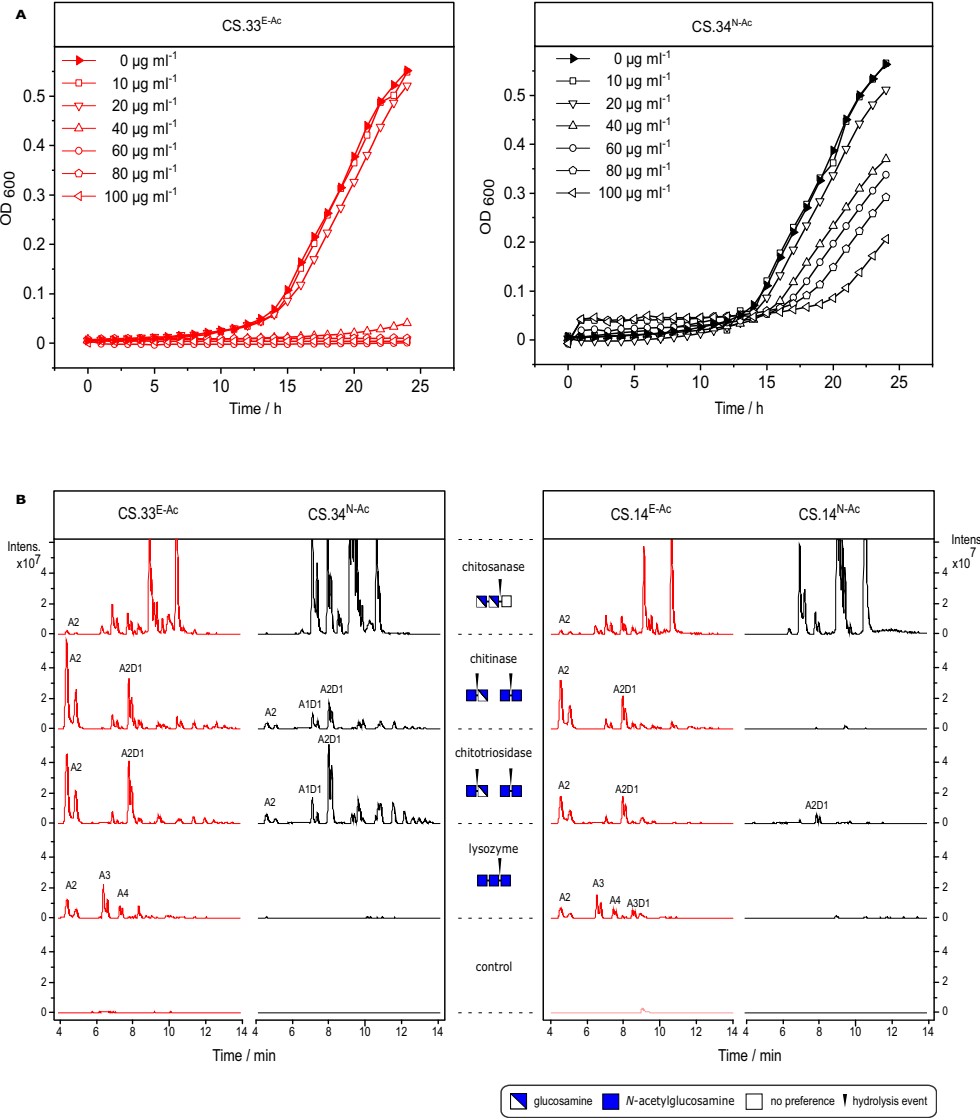

**Fig. 5 | PA influences the antimicrobial activity and enzymatic degradability of chitosans. A** Growth of *Pseudomonas syringae* pv. *tomato* (DC3000 [pVSP61]) in the presence of different concentrations of enzymatically (left panel) and chemically (right panel) *N*-acetylated chitosan polymers, measured as OD600 over 24 h. **B** UHPLC-ESI-MS analysis, showing base peak chromatograms of the oligomeric hydrolysis products (A = GlcNAc, D = GlcN) of enzymatically (left panel) and chemically (right panel) *N*-acetylated chitosan polymers of $F_A \approx 0.3$ (left panel) or $F_A \approx 0.1$ (right panel) after 24 h of incubation with chitinase (ChiB), chitosanase (Csn174), human lysozyme, or human chitotriosidase, or without an enzyme. The subsite specificities of the chitinolytic enzymes are indicated in the central gutter.

blocks to confer antibacterial activity even at the rather high $F_A$ of 0.33, where a random-PA chitosan would have an average GlcN block size of only 3. The antibacterial activity of chitosans against Gram-positive bacteria appears to rely on their interaction with teichoic acids[10,53] which may be less influenced by PA. Both the biotech and conventional chitosans showed no cytotoxicity toward human keratinocytes (HaCaT) or primary human umbilical vein endothelial cells (HUVEC) (Supplementary Fig. 6). The most striking difference between the two types of chitosan was their sensitivity to different chitosanolytic enzymes (Fig. 5b). Incubation of the random-PA chitosan with a bacterial chitosanase that strongly favours GlcN units close to its cleavage site[21,54] predominantly yielded fully deacetylated and partially acetylated dimers and trimers. In contrast, the biotech chitosan yielded almost exclusively fully deacetylated dimers and trimers, consistent with a block-PA organisation (GlcN blocks are degraded to $GlcN_2$ and $GlcN_3$, whereas GlcNAc blocks are not degraded). Conversely, incubation of the random-PA chitosan with a bacterial chitinase that strongly favours GlcNAc units[19,20,55] predominantly yielded partially

acetylated dimers and trimers but almost no fully-acetylated chitobiose. In contrast, the biotech chitosan predominantly yielded chitobiose $GlcNAc_2$ derived from the GlcNAc blocks. A similar behaviour was observed using the human enzyme chitotriosidase, which has subsite preferences resembling the bacterial chitinase[56,57]. Human lysozyme, which has an almost absolute specificity for GlcNAc units close to its cleavage site[58,59], did not produce oligomers from the random-PA substrate but generated fully-acetylated dimers to tetramers from the block-PA substrate. Similar results were obtained when the enzymatic degradation products of two chitosans with an even lower $F_A$ of ca. 0.1 were compared. As anticipated given the low $F_A$, chitosanase generated similar products with both substrates, but chitinase and chitotriosidase yielded chitobiose alone from the block-PA chitosan. Surprisingly, lysozyme digestion of the block-PA chitosan yielded fully-acetylated GlcNAc oligomers up to DP = 4, indicating the frequent presence of heptameric or even longer GlcNAc blocks in this polymer[60]. With a random PA, the anticipated frequency of heptameric GlcNAc blocks would be $0.1^7$, equivalent to one in every 10 million

monomeric units, or one in every 10,000 polymers with a DP of 1000. As expected, thus, lysozyme did not yield any oligomeric products from the random-PA chitosan with $F_A$ of ca. 0.1.

The influence of PA on the interaction of chitosans and chitinases can be regarded as an example for the more general interaction of chitosans and proteins. Other examples would be sequence-dependent bacterial or fungal chitosanases[21] and chitinosanase[61], or other enzymes such as matrix metalloprotease 2 (MMP2)[62]. Similarly, interactions with other proteins such as the chitin receptors CERK1 in plants[23–25] and TLR2 in human cells[63] can be expected or have even already been shown to be pattern-specific. And beyond receptors, the binding of chitosans to glycoproteins and non-proteinaceous targets such as mucins[64] or glycosaminoglycans[65] can be assumed to be strongly influenced by PA.

While the above experiments featured chemically and enzymatically N-acetylated chitosans, almost all commercial chitosans are prepared by the partial de-acetylation of chitin rather than the partial N-acetylation of polyglucosamine. We therefore selected the two enzymatically N-acetylated chitosans described above (the first $F_A = 0.33$ and DP = 800; the second $F_A = 0.14$ and DP = 200) as well as two commercial, chemically deacetylated pharmaceutical-grade chitosans with similar parameters (the first $F_A = 0.24$ and DP = 1300; the second $F_A = 0.17$ and DP = 200) for more detailed comparative analysis (Fig. 6). The two commercial chitosans have been used in biomedical research, especially for the transfection of human cells[66–68]. We, therefore, prepared polyplexes by mixing these chitosans with plasmid DNA encoding eGFP at different molar charge chitosan/DNA ratios (Supplementary Fig. 7a). For most ratios, we obtained particles with a diameter of 200–300 nm, although the high-$F_A$ high-DP conventional chitosan generated larger particles at higher chitosan/DNA ratios (Fig. 6a; Supplementary Fig. 7b). The dispersity in size of the polyplexes was similarly low when using either of the two chitosans (Supplementary Fig. 7c), thus suggesting that in both cases, the formation of complexes proceeded under a similar route yielding a single population of particles. All subsequent experiments were performed at a $NH_3^+/PO_4^-$ molar charge ratio of 8, which produced stable particles (Supplementary Fig. 7c). Chitosan/DNA polyplexes tend to aggregate at neutral pH because they lose their surface charge, thus their stability in physiological media is critical for their suitability as a transfection reagent[69]. Given that in previous studies[70,71] we had found that the colloidal stability of chitosan nanoparticles and nanocapsules strongly depended on $F_A$ and, to a lesser extent, on DP, but not on the physiological medium used (RPMI-1640, ECGM, OptiMEM), we limited the stability assays to OptiMEM as the medium used in subsequent transfection studies. With the exception of the low-$F_A$ low-DP conventional chitosan, all polyplexes were stable in OptiMEM for 4 h (Fig. 6b). When these polyplexes were used to transfect MCF7 breast cancer cells, only the low-$F_A$ low-DP biotech chitosan achieved significant transfection efficiency on par with the positive control, Lipofectamine (Fig. 6c). Importantly, both biotech chitosans were degraded by human lysozyme, as shown by the production of chitobiose and chitotriose, whereas the conventional chitosans were not (Supplementary Fig. 7d).

## Discussion

Chitosan is a versatile biopolymer with useful biological activities, but applications have been held back by the poorly understood structure-function relationships of partially acetylated chitosans[5,72,73]. Recent improvements in analytical techniques and reproducible protocols for the synthesis of well-characterised second-generation chitosans with known DP and $F_A$ have provided insight into these molecules[11,74]. However, all commercial chitosans are produced chemically and are thought to have a random PA[15], so it is not yet possible to investigate the influence of this property on chitosan behaviour. We have now demonstrated that recombinant CDAs can be used to N-acetylate

polyglucosamine in the presence of excess acetate, yielding chitosan polymers with unique, non-random PAs. Two enzymatically generated chitosans featuring a block-wise distribution of acetyl groups were compared in detail to chemically generated random-PA chitosans with a similar DP and $F_A$. This revealed the strong influence of acetylation patterns on both the physicochemical properties and biological activity of the chitosans. The availability of these closer-to-nature third-generation biotech chitosans opens a new dimension in chitosan research and exploitation by allowing the control of all three key parameters (DP, $F_A$, and PA).

Chitosans are binary copolymers of partly hydrophobic GlcNAc and hydrophilic GlcN subunits. In conventional chitosans, these monosaccharide units appear to be randomly distributed and the molecules therefore follow the general law of behaviour in aqueous solution described previously[75]. At low $F_A$ (≤0.25), they show polyelectrolyte behaviour, whereas at higher $F_A$ (≥0.5), they behave as typical hydrophobic polymers. At intermediate values, their behaviour is governed by the balance between hydrophilic and hydrophobic interactions. Enzymatic N-acetylation provides access to biotech chitosans with non-random acetylation patterns that clearly behave in a different manner to conventional chitosans in solution. For example, block-PA chitosans possess the properties of both polyelectrolytes and hydrophobic polymers in a single molecule, potentially leading to phenomena such as microphase separation as reported for other biopolymers. Indeed, an influence of the pattern of substitution on the physicochemical properties (e.g., solubility, phase behaviour, solution rheology, gelation, crystallinity) and conformation of several polysaccharide families has been well documented. For example, the solution properties (such as gelling potential) of partially methyl esterified homogalacturonan pectins strongly depend on both the degree and pattern of methyl esterification[76–78]. In alginates, the intrinsic viscosity[79] and the gel strength of calcium alginate gels correlate with the block composition[80]. In galactomannans, in silico modelling has allowed investigating how the distribution of the galactose side chains influences the conformation and fundamental properties of chains with random, block, or alternate substitution patterns. These studies predicted that for the same degree of substitution, the block polymer is stiffer than the random and alternate polymers[81], in concordance with available experimental data for galactomannans and with our experimental results on random and block-wise patterned chitosans. The pattern of substitution of galactomannans is also known to influence the interaction properties with xanthan gum[82]. In the same way, we can safely predict that the distribution of more hydrophobic versus charged domains within chitosan polymers will govern their interactions with other, similarly heterogeneous but oppositely charged polymers, such as pectins[83], alginates[84], dextran sulphate[85] or mucins[3]. In the future, polyelectrolyte complexes of chitosans and e.g. pectins with defined patterns of acetylation and methyl esterification, respectively, could facilitate the targeted, sustained, and controlled release of hydrophobic and hydrophilic drugs as well as vaccines and genes.

The biological functions of copolymers and their oligomeric degradation products are strongly influenced by their degree and pattern of substitution, as reported for pectin methyl esterification[86] and glycosaminoglycan sulfation[87]. Regarding chitosans, a crucial influence of the $F_A$ on biological activities has been proven for both polymers and oligomers[5,6,8,10,18,72,73], while a role of the PA was only recently revealed for chitosan oligomers[22,23,31,88], and has not been reported previously for chitosan polymers. In part, this reflects direct physicochemical interactions with target molecules or structures. For example, block-PA chitosans may combine the antimicrobial activity of low-$F_A$ chitosans[5,51] with the ability of high-$F_A$ chitosans to induce pathogen resistance in plants[5,8], allowing the development of dual-function plant protection products. The pattern of substitution also influences the interactions with enzymes, particularly with

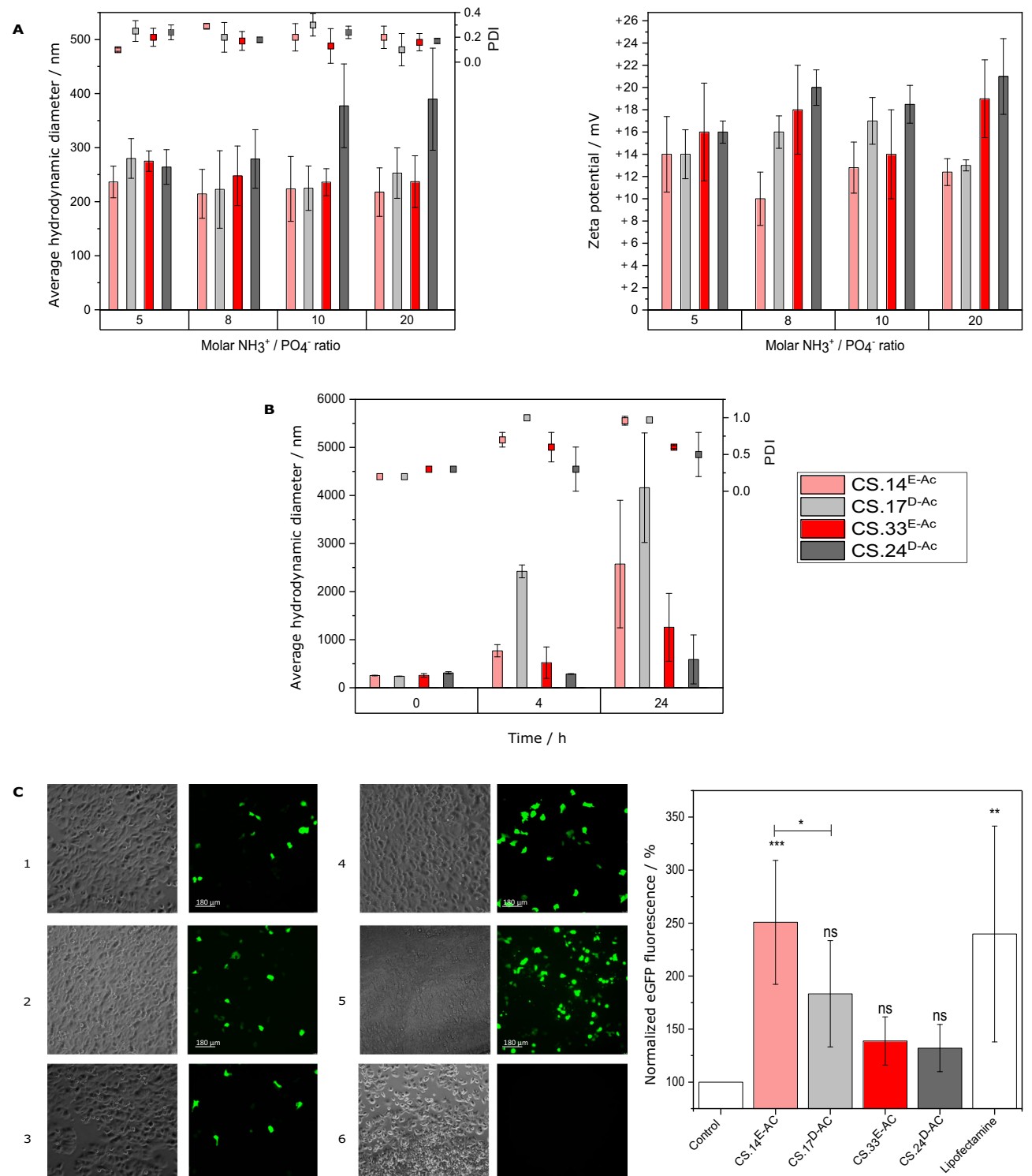

**Fig. 6 | PA influences the transfection efficiency of chitosan/DNA polyplexes.**
**A** Physicochemical characteristics of polyelectrolyte complexes formed from enzymatically *N*-acetylated and chemically deacetylated chitosan polymers ($F_A \approx$ 0.3 or 0.1) and plasmid DNA (pDNA) at different $NH_3^+/PO_4^-$ molar charge ratios: Z-average hydrodynamic diameter and PDI (left panel) and zeta potential (right panel) as determined by dynamic light scattering. Data represent three independent experiments plotted as means ± SD. **B** Stability of chitosan/pDNA polyplexes in transfection media: the polyplexes were formed at a $NH_3^+/PO_4^-$ molar charge ratio of 8 and were incubated for different times in OptiMEM at pH 6.8 and 37 °C. Data represent three independent experiments plotted as means ± SD. **C** Transfection efficiency of the chitosan/pDNA polyplexes: Representative phase contrast and fluorescence microscopy images (scale bar 180 μm) of MCF7 cells transfected with chitosan/pDNA polyplexes or Lipofectamine-pDNA as a positive control, showing GFP expression 48 h post-transfection: 1 = CS.33[E-Ac], 2 = CS.24[D-Ac], 3 = CS.17[D-Ac], 4 = Lipofectamine 2000, 5 = CS.14[E-Ac], 6 = control cells (left panel); fluorescence intensity data normalized to negative control of MCF7 cells transfected with carriers and 2.5 μg pDNA per well after 48 h of incubation; negative control = cells not transfected; positive control = cells transfected with Lipofectamine 2000 (right panel). Data represent three independent experiments plotted as means ± SD. Statistical test: one-way ANOVA followed by Tukey's multiple comparisons test; horizontal bar represents significant difference between treatments (*$p < 0.05$; **$p < 0.01$; ***$p < 0.001$; ns = not significant compared to negative control).

depolymerases such as chitinases and chitosanases[21,27], poly-galacturonases and pectin/pectate lyases[89], or heparinases and heparan sulphate lyases[90,91]. We have shown that the PA strongly influences the sensitivity of chitosan polymers to different enzymes, affecting the kinetics, nature, and quantities of the resulting oligomeric products. This is important because chitosan polymers are typically degraded in target tissues by sequence-dependent chitosan hydrolases, potentially giving rise to biologically active chitosan oligomers with a specific architecture[31,57,88,92]. Biodegradability is also a prerequisite for the biocompatibility of chitosans used as pharmacological excipients. Chitosan nanoparticles and nanocapsules are ideally suited for the transmembrane delivery of drugs, vaccines, and genes[38,93], but the inability of human chitinases to break down the most efficient, low-$F_A$ chitosans precludes their application in medicine[57,92]. Enzymatically $N$-acetylated low-$F_A$ block-PA chitosans could offer a game-changing innovation in this scenario.

Depending on the enzyme used for $N$-acetylation, chitosans can be generated with either more block-wise acetyl groups or more regular distribution of GlcNAc. Like chitinases and chitosanases[21], chitosan deacetylases feature a substrate-binding cleft consisting of several subsites, each binding a single monomeric subunit[26] and probably showing unique preferences or specificities for binding GlcN or GlcNAc units. The fungal enzymes used in this study are each likely to possess four subsites, ranging from {−2} to {+1}, with subsite {0} binding the GlcNAc unit that is deacetylated to form GlcN. Most CDAs prefer GlcNAc at all subsites, but CnCDA4 prefers GlcN at subsite {−1} and is therefore classed as a chitosan deacetylase rather than a chitin deacetylase[88]. A typical chitin deacetylase, such as PesCDA or AnCDA, prefers GlcNAc at subsite {−1} and tends to add acetyl residues downstream (further towards the reducing end) of existing GlcNAc units, thus creating GlcNAc blocks with interspersed GlcN blocks and favouring a block-PA architecture. In contrast, the chitosan deacetylase CnCDA4 tends to add acetyl groups downstream of a GlcN unit, generating chitosans with more regularly distributed GlcNAc units. PgtCDA appears to have only a slight preference for GlcN at subsite {−1}[32], probably explaining why it yields a chitosan with an almost random PA. The varying preferences at subsites {−2} and {+1} also influence the distribution of acetyl groups on the product, so that each enzyme produces chitosans with a unique PA. In this study, we investigated four closely related fungal CDAs, and the products differed for each enzyme. Given the huge diversity of naturally occurring CDAs (not only in fungi, but also in bacteria, insects, and even viruses) and the possibilities offered by protein engineering to modify subsite specificities or preferences, or alter processivity, the enzymatic $N$-acetylation of polyglucosamine will allow the unlimited production of third generation chitosans for research and development. Given that chitosans are the only polycationic counterparts to many different polyanionic biopolymers, the impact of this innovation will extend far beyond chitosan research into many aspects of advanced functional materials, medicine, agriculture, and industrial biotechnology.

In conclusion, we here report on the biotechnological production and characterisation of chitosan polymers with non-random acetylation patterns, using recombinant chitin deacetylases acting in reverse mode on polyglucosamines, and their potential as advanced functional biomaterials. In contrast to all conventional chemically produced chitosans which possess random PA, the PA of the biotech chitosans ranges from large block-like structures to near-even distribution, depending on the enzyme used. These biotech chitosans which differ from their conventional counterparts in terms of physicochemical properties and biological activities could become the third generation of chitosans, structurally controlled in all three key parameters. While in the present study, we provide evidence for the crucial role of PA in determining the properties and functionalities of chitosans, more specifically dedicated follow-up studies will expand this new dimension of chitosan research in the future. As an example concerning

physicochemical properties, the influence of PA on the behaviour of these chitosans in other physically, enzymatically, or other covalently crosslinked chitosan gel systems and on the mechanical, swelling, and diffusion properties of such gels as well as of their suitability for cell biology and tissue engineering will be exciting fields of further studies. Concerning biological activities, it will e.g. be interesting to investigate their potential cytotoxicity or biocompatibility towards other cells and tissues, including e.g. cancer cell lines, and their antimicrobial activities against other bacteria, including human pathogenic ones, and other microorganisms such as fungi and oomycetes. Another promising area for future research is the role of PA in the interaction of chitosans with proteins such as pattern recognition receptors in human, animal, or plant immune systems, and chitin or chitosan modifying enzymes. Perhaps most importantly, we here describe a chitosan polymer with a block-like distribution of acetyl groups that is easily degraded by human chitinases and even lysozyme, despite its low $F_A$. Biodegradability in humans will facilitate the use of chitosan nanostructures for the delivery of drugs, genes, and vaccines, which is not possible with current chitosans. We show that this block-PA chitosan can form stable polyelectrolyte nanocomplexes with nucleic acids, achieving better transfection efficiency than conventional chitosans. These offer game-changing potential in many fields, including the development of reliable and stable RNA-based vaccines.

## Methods

### Production and purification of enzymes

**Chitosan hydrolases.** The hydrolytic enzymes required for EMS fingerprinting chitinase ChiB (from *Serratia marcescens*, Uniprot acc. no. A0A059UJT0[94] and chitosanase Csn174 (from *Streptomyces* sp. *N174*, Uniprot acc. no. P33665[54] were produced heterologously in *E. coli* as fusion proteins with *Strep*-tag II and purified using Streptactin affinity chromatography. Chitinosanase (from *Alternaria alternata*) was purified from the spent medium of the fungus using cation exchange chromatography[61].

The chitosanolytic enzymes used to assess enzymatic digestibility of chitosan polymers were either produced heterologously in *E. coli* and purified as described above (chitinase ChiB; chitosanase Csn174) or expressed homologously in HEK293 cells (chitotriosidase ChT, Uniprot acc. no. Q13231[88] as fusion protein with His6-tag, and purified using Ni-NTA affinity chromatography.

**Chitin deacetylases.** Fungal chitin deacetylases were produced recombinantly in *E. coli* and purified using Streptactin affinity chromatography. PesCDA (from *Pestalotiopsis* spec., GenBank acc. no. KY024221[31]), PgtCDA (from *Puccinia graminis* f.sp. *triciti*, NCBI acc. no. XP_003323413.1[32], CnCDA4 (from *Cryptococcus neoformans*, Uniprot acc. no. Q96TR5[33] were produced as previously described. AnCDA was heterologously expressed in *E. coli* Rosetta 2 (DE3) as a truncated version (Δ1-19, proposed signal peptide) of the chitin deacetylase An12g04480 from *A. niger* CBS 513.88. A C-terminal *Strep*-tag II and SerAla linker (SA-WSHPQFEK) was added for protein purification.

### Preparation of chitosans

**Chemical de-acetylation.** Chemically deacetylated chitosans were obtained commercially from Heppe Medical Chitosan GmbH (HMC⁺, Halle/Saale, Germany), namely ultrapure biomedical grade HMC 70/5 (Batch No. 212-170614-01) and HMC 70/100 (Batch No. 212-170114-01). They were produced from snow crab shell α-chitin using a heterogeneous de-$N$-acetylation process in hot alkali[95].

**Chemical $N$-acetylation.** Chemically $N$-acetylated chitosans were prepared using acetic anhydride (Roth, Germany)[74]. Briefly, chitosan (DP 1300, Đ 1.8, $F_A$ 0.03; prepared from shrimp shell α-chitin by four sequential heterogeneous de-$N$-acetylation steps using hot alkali by Mahtani Chitosan, Veraval, India[95] was solubilised in water by adding

5% stoichiometric excess of acetic acid and stirred until completely dissolved. One volume of 1,2-propanediol (Roth, Germany) was added to the chitosan solution to reduce the isoelectric constant of the medium and to help chitosan chains to adopt an open conformation. Acetic anhydride was then added in the required molar amount to reach the target $F_A$. After 24 h at RT, chitosan was precipitated with an ammonia solution (23% w/v (Roth, Germany)). The polymer precipitate was washed to neutrality and freeze-dried for subsequent use.

**Enzymatic N-acetylation.** Enzymatically N-acetylated chitosans were produced using different fungal chitin deacetylases recombinantly produced in *E. coli* as described above. The same chitosan polymer as described above for chemical N-acetylation was used as a starting material and dissolved in 5% stoichiometric excess of acetic acid to achieve complete dissolution. This solution was further diluted 1:2 in 3 M sodium acetate (Roth, Germany) pH 7.5 to a final chitosan concentration of 1 mg mL$^{-1}$. The small-scale (2 mL) time series experiments were performed for 24 h at 37 °C using the following final concentrations of enzymes: AnCDA, 1.2 μM; CnCDA4, 700 nM; PesCDA, 175 nM; PgtCDA, 600 nM. Small volumes (1–5 μL) of concentrated stock solutions of enzymes were used to prevent dilution of the reaction buffer. Samples were taken at different time points to determine the $F_A$ by EMS fingerprinting as described below. The large-scale (3.5 L) production using PesCDA was performed under the same conditions as described above, the $F_A$ was monitored using EMS fingerprinting and additional enzyme was added until the desired $F_A$ was reached. Then, chitosan was precipitated using acetone (1:1 v:v), the precipitate was collected by centrifugation (20 min, 12,000 × g, 4 °C), washed three times with water adjusted to pH 9 using ammonia, and then three more times with distilled water, before being freeze dried. To ensure complete removal of salts remaining from high salt conditions required for enzymatic N-acetylation, samples were re-dissolved in water and dialysis was performed using 12 kDa cut-off dialysis membranes (Repligen, Ravensburg, Germany). Samples were again freeze-dried before being used for structural and functional analysis.

**Structural analysis of chitosans**
Chitosan polymers were structurally analysed using standard methods, as described below. The values obtained for all chitosans used in this study are given in Supplementary Table 2.

The number and weight-average DP of chitosan polymers were determined using HPSEC-RID-MALLS[75,96], using TSKgel® columns (PW$_{XL}$-CP-guard column + G6000 PW$_{XL}$-CP + G5000 PW$_{XL}$-CP + G3000 PW$_{XL}$-CP) (Tosoh, Griesheim, Germany) and degassed ammonium acetate (Roth, Germany) buffer (0.15 M, pH 4.5) as an eluent at a flow rate of 0.5 mL min$^{-1}$. Light scattering intensity measurements were performed to determine the Mw and Mn following the Rayleigh-Debye equation (using WinGPC UniChrom sofwater, PSS, Germany); these values were used to calculate the molecular weight dispersity (Đ).

The average $F_A$ of chitosan polymers was determined either using $^1$H-NMR[97] recording 200–250 spectra on an AV300 or DPX300 300 MHz spectrometer (Bruker, USA), or using chitinase/chitosanase-mass spectrometric fingerprinting[98].

The PA of chitosan polymers was determined either using $^{13}$C-NMR dyad analysis[15] or using chitinosanase-mass spectrometric fingerprinting[27,96]. For $^{13}$C-NMR analysis, 100 mg of purified chitosan was dissolved in 10 mL of 0.07 M HCl (Roth, Germany) and stirred overnight at room temperature. The solution was treated with 10 mg sodium nitrite (NaNO$_2$ (Roth, Germany)) for partial depolymerisation, stirred for 4 h, and subsequently freeze dried. Samples were dissolved in 1 mL acidic solution of D$_2$O (1 ml 99.9% D$_2$O, 5 μl DCl (Sigma-Aldrich, Germany) and freeze-dried. Finally, samples were dissolved in D$_2$O (Roth, Germany), and $^{13}$C-NMR spectra were recorded on a 600 MHz DD2 instrument (Agilent, USA). The dyad frequencies of chitosan samples were

determined based on the C-5 resonance region, and deviation from random statistics (P$_\Sigma$) was analysed[27] based on former work by Varum et al. and Weinhold et al.[15,16,99,100]. For chitinosanase-mass spectrometric fingerprinting[96], chitosan samples were hydrolysed with purified chitinosanase[61] (see above) using the following conditions: 1 mg mL$^{-1}$ chitosan, 3.5 μg mL$^{-1}$ chitinosanase, 200 mM ammonium acetate buffer, pH 4.2, incubation at 37 °C for two days. After one day of incubation, chitinosanase concentration was increased to 6.8 μg mL$^{-1}$. Oligomeric products were analysed using semi-quantitative HILIC-ESI-MS[61,96]. Normalised mass-fractions of the hydrolysis products with DP 2–10 were calculated. Based on the DA/XX cleavage preference of the chitinosanase, we calculated the frequencies of block sizes represented by the detected hydrolysis products. Weight-average block sizes and differences of block size frequencies between chemically and enzymatically acetylated chitosans were calculated as well.

**Physicochemical solution properties of chitosans**
The dynamic viscosity of chitosan aqueous solutions, solubilised with 5% stoichiometric excess of acetic acid and 0.1 M NaCl was measured using an AMVn automated rolling ball microviscometer (Anton Paar, Ostfildern, Germany), using a capillary of 1.6-mm diameter at an angle of 40° and at 25 °C. The dynamic viscosity was calculated from the average of four runs, either in water containing 5% stoichiometric excess of acetic acid or in 0.1 M NaCl. From the relative viscosity $\eta_{rel}$ thus determined, the specific viscosity $\eta_{sp}$ ($\eta_{sp} = \eta_{rel} - 1$) was calculated by joint extrapolation to zero concentration of the Huggins, Kraemer, and single point relationships[101].

Chitosan-TPP (Sigma-Aldrich, Germany) nanoparticles were prepared using the ionic gelation process[36,37]. A series of NH2/TPP molar ratios (0.3, 0.6, 0.9, 1.2, 1.5, 1.8, 2.1, 2.4, 2.7, and 3) were screened to test the formation of particles. For this, stock solutions of chitosans and TPP were prepared at 2 mg mL$^{-1}$ (filtered using 0.45-μm filter) and 7 mg mL$^{-1}$ (filtered using 0.22 μm filter), respectively. Chitosan-TPP particles were generated spontaneously upon dropwise addition of TPP into the chitosan solution stirring at 750 rpm and at room temperature. All particles were prepared at a chitosan:TPP volume ratio of 3:1. The resulting particles were characterised for their size, polydispersity index (PDI), and derived count rate (DCR) using dynamic light scattering with non-invasive back scattering (DLS-NIBS) at a measurement angle of 173° using the method of cumulants. The zeta potential was measured by mixed laser Doppler velocimetry and phase analysis light scattering (M3-PALS). A Malvern Zetasizer NanoZS (Malvern Panalytical, Malvern, UK) fitted with a red laser ($\lambda = 632.8$ nm) was used for both analyses. The Zetasizer Software (v 7.12, Malvern Panalytical) was used to acquire and evaluate the data.

Nanocapsules were prepared by the solvent displacement technique[102], with some modifications. Briefly, an organic phase was formed by dissolving 40 mg of lecithin (Epikuron 145 V, Cargill Deutschland GmbH & Co. KG, Hamburg, Germany) in 1 mL of ethanol, followed by the addition of 125 μL of Miglyol® 812 (Sasol GmbH, Witten, Germany) and adding ethanol up to 10 mL. This organic phase was immediately poured over 20 mL of the aqueous phase composed of a chitosan solution (0.5 mg mL$^{-1}$ dissolved in water with 5% stoichiometric excess of acetic acid). Nanocapsules were formed spontaneously due to the organic solvent's diffusion and Marangoni effects of the organic phase[103]. Finally, the ethanol and some of the water were evaporated at 40 °C under vacuum on a R-210 Rotavapor (Büchi Labortechnik, Essen, Germany) and the volume of the formulations was reduced to 10 mL. Nanocapsules were characterised on the basis of average size distribution, PDI, DCR, and zeta potential, as described above.

To prepare chitosan nanoparticles by electrospraying, chitosans were dissolved in 30% acetic acid and 30% ethanol at a concentration

of 5 mg mL$^{-1}$[40]. For the preparation of the solutions, chitosan samples were first dispersed in water, followed by the addition of acetic acid and ethanol. Mixtures were stirred overnight prior to the electrospraying process. Chitosan solutions were electrosprayed using a high voltage generator (ES50P-10W, Gamma High Voltage Research, Ormond Beach, FL, USA) and a syringe pump (New Era Pump Systems, Farmingdale, NJ, USA) to provide specific voltages and solution flow rates, respectively. For scanning electron microscopy (SEM) to investigate the morphology of the particles, the samples were attached on metal stubs with double-sided adhesive carbon tape and coated with 6 nm of gold for better conductivity using a sputter coater (Leica Coater ACE 200, Leica, Vienna, Austria), prior to visualisation using a Quanta FEG 3D SEM (FEI, Eindhoven, The Netherlands). The average particle size (50 particles per image) was determined using ImageJ (version 1.5).

The rheological properties of chitosans were measured at a concentration of 30 mg mL$^{-1}$ (dissolved in 5% stoichiometric excess of acetic acid) using a Kinexus Ultra rheometer (Malvern Panalytical, Malvern, UK). A cone plate model (CP 4/40, PL 65) was used to monitor the storage and loss modulus. Frequency sweep measurements were performed where the frequency varied between 0.01 and 100 Hz (0.0628-62.8 rad s$^{-1}$) and the strain was 20% (within the linear viscoelastic region). Shear viscosity studies were performed at a shear rate of 0.01–10 s$^{-1}$. All samples were measured at 25 °C. As the rheological measurements required relatively high amounts of chitosan which were not available for the chemically N-acetylated chitosan of $F_A$ 0.30 and DP 700 that was used for all other experiments, a chitosan of $F_A$ 0.30 and DP 1700 was used for the rheology experiments. Even when the DP does influence the overall magnitude of the viscoelastic and steady-shear viscosity parameters, the comparison between the two polymers of varying DP is valid as long as both solutions are in the entangled regime[37,41].

For gelation studies, the required amount of the dry chitosan was dissolved in 0.5 M acetate buffer pH 4.5 by accurate weighing at a concentration of ca. 47 mg mL$^{-1}$ and was left to fully dissolve under moderate magnetic stirring during 3 days. Once fully dissolved, the stock solutions were kept under refrigeration until further use. Genipin (Challenge Bioproducs Co., Taiwan, PRC) stock solution (100 mg mL$^{-1}$) was prepared by dissolving the crystalline powder in ethanol and was freshly made ahead of use. Chitosan solutions (ca. 4–5 mL) were made in capped glass vials by diluting the stock solutions to ca. 15 mg mL$^{-1}$, fully homogenised using a vortex and equilibrated to 40 °C in a water bath. Before loading to the rheometer, an aliquot of the genipin stock solution was added to achieve a genipin/GlcN molar ratio of 0.5. The final concentration of chitosan was ca. 14 mg mL$^{-1}$. The amount of added ethanol was 6.8% (v/v). The gelation studies were performed on an Anton Paar GmbH (Graz, Austria) MCR 302 rheometer equipped with a stainless-steel parallel plate geometry (I-PP60/SS/CX) coated with PTFE (diameter 60 mm) and using a gap of 2 mm. The rheometer was connected with a water bath in combination with a Peltier heating system for temperature control set to 40 °C. A layer of silicone oil with a high viscosity was applied to the annulus of the measuring geometry to avoid dehydration of the sample during the experiment. The instrument was controlled with RheoCompass$^{TM}$ software v1.24. The measuring tests followed the sequence: (i) steady-shear flow (0–100 s$^{-1}$); (ii) oscillatory time sweep multiwave measurements using a fundamental wave frequency of 1 rad s$^{-1}$ and strain of 5%; a total of nine harmonics (3, 5, 7, 9, 11, 13, 15, 17, and 19 rad s$^{-1}$) were set; measurements were recorded at one-minute intervals; the time between the addition of genipin to the chitosan solution and the start of the oscillatory tests was chronometred with a smartphone and used to adjust the measurement times; (iii) amplitude (strain) sweep at a frequency of 1 rad s$^{-1}$ that ensured that all measurements in (i) and (ii) were conducted within the linear viscoelastic region. The remaining of

the chitosan solutions with added genipin was kept at 40 °C in the water bath for visual observation.

CD spectroscopy was performed using a Chirascan Plus CD spectrophotometer (Applied Photophysics, Surrey, UK) with a LAAPD detector and Chirascan Spectrometer Control Panel software version 4.4 (Applied Photophysics). Far-UV CD analysis was performed from 180 to 260 nm with a 0.5 nm step size. Chitosan samples were dissolved in stoichiometric excess of acetic acid at concentrations of 0.5 mg/mL. All measurements were performed at 25 °C using a 0.1 mm precision cuvette (Hellma, Müllheim, Germany); each sample was scanned ten times and results were averaged, no smoothing was used. The sample solvent was also scanned under identical conditions and subtracted from the sample spectra.

Pyrene was selected as a hydrophobic fluorescent probe to determine the hydrophobic domains within the chitosan samples. Pyrene (Sigma-Aldrich, Germany) dissolved in methanol was added to chitosan solutions (0.625, 1.25, 2.5, and 5.0 mg mL$^{-1}$ in 100 mM NaCl) to give a final concentration of 2 μM. Fluorescence spectra were taken on a Jasco FP-6500 spectrofluorometer (Jasco, Pfungstadt, Germany) at 25 °C. The excitation wavelength was fixed to 343 nm, and emission spectra were recorded between 360 and 550 nm; the ratio between the peak intensities of the first peak at 374 nm ($I_1$) and of the third peak at 385 nm ($I_3$), which vary based on the hydrophobicity of the sample, was plotted[104,105].

## Biological functionalities of chitosans

In vitro antibacterial activity of chitosans was tested against the Gram-negative bacterium *Pseudomonas syringae* pv. *tomato* (DC3000 [pVSP61]) provided by MPI, Cologne, Germany, and against the Gram-positive film-forming bacterium *Bacillus licheniformis*. These strains were selected because the respective bioassays to quantify bacterial growth are well established in our lab[106]. A pre-culture of *P. syringae* was grown in NYG medium (0.5% (w/v) peptone, 0.3% (w/v) yeast extract, 2% (v/v) glycerol, pH 5.5; all ingredients were purchased from Roth, Germany) at 30 °C under agitation at 100 rpm for two days. Antibacterial assay was performed in a 96-well plate by mixing 40 μL of chitosan to 160 μL of bacterial suspension with OD = 0.0125 or medium as a blank, yielding final chitosan concentrations of 0, 10, 20, 40, 60, 80, or 100 μg mL$^{-1}$. Growth of bacteria was measured continuously for 24 h at an interval of 10 min, by measuring the optical density at $\lambda$ = 600 nm (OD$_{600}$) using a UV/Vis microplate reader (SpectraMax M2, Molecular Devices, Sunnyvale, CA, USA) at 26 °C. A pre-culture of *B. licheniformis* was grown in LB medium (1% (w/v) NaCl, 1% (w/v) peptone, 0.5% yeast extract, pH 7) at 37 °C under agitation at 250 rpm overnight. Antibacterial assay was performed in a 24-well flat bottom deep well plate by mixing 50 μL of bacterial suspension or medium as a blank, with 1 mL of chitosan solution and 4 mL of MM1 P100 medium (3% (w/v) glucose, 0.13% (w/v) MgSO$_4$ 7 H$_2$O, 0.17% (w/v) KH$_2$PO$_4$, 0.005% (w/v) CaCl$_2$ 2 H$_2$O, 0.2% (v/v) RPMI-1640 vitamins solution (Sigma-Aldrich, Germany), 0.1% (v/v) trace element solution containing 0.25 (w/v) FeSO$_4$ 7 H$_2$O, 0.21% (w/v) C$_4$H$_4$Na$_2$O$_6$ 2 H$_2$O, 0.18% (w/v) MnCl$_2$ 4 H$_2$O, 0.015% (w/v) CoCl$_2$ 6 H$_2$O, 0.003% (w/v) CuSO$_4$ 7 H$_2$O, 0,026% (w/v) H$_3$BO$_3$, 0.002% (w/v) Na$_2$MoO$_4$, 0.0021% (w/v) ZnCl$_2$, pH 6.8) to induce biofilm formation[107], yielding final chitosan concentrations of 0, 10, 20, 30, 40, 50, 75, or 100 μg mL$^{-1}$. After overnight incubation at 37 °C without shaking, 20 mL per well were diluted with 180 μL of MM1 P100 medium in a separate 96-well plate, and the OD was measured at $\lambda$ = 600 nm (OD$_{600}$) using a UV/Vis microplate reader (Epoch 2 Microplate Spectrophotometer, Agilent Technologies, Santa Clara, CA, USA). In both cases, chitosan stock solutions were prepared in 5% stoichiometric excess of acetic acid.

The in vitro cytotoxicity of chitosans was studied using the MTT (Fluka, Germany) assay on HaCaT cells as a model cell line (obtained from the dermatological clinic at the University Hospital in Münster,

Germany) and on HUVECs as model primary cells (obtained from the Institute of Physiological Chemistry and Pathobiochemisty at the University Hospital in Münster, Germany). In both cases, a cell suspension (100 µl containing ca. $10^4$ cells per well) was seeded into a 96-well tissue culture plate and incubated for 24 h, cells were then washed twice with PBS (Merck, Germany) before addition of samples at varying concentrations and further incubation for 24 h. Samples were then removed and replaced by 100 µl medium containing 25 µl of MTT solution (5 mg mL$^{-1}$ in PBS). Following incubation for 3 h, the medium was replaced by 100 µl of DMSO (Sigma-Aldrich, Germany) and the plate was shaken for 10 min at 300 rpm. Absorbance was measured at $\lambda = 570$ nm using a UV/Vis microplate reader (Multiscan GO 60, Thermo Fisher Scientific, Waltham, MA, USA). Relative viability was calculated in percent of the OD value of cells growing in the absence of chitosan. Triton X-100 (Sigma-Aldrich, St. Louis, MO, USA) in PBS (4%) was used as a positive control.

In vitro degradation of chitosans was performed using four different enzymes—chitinase chiB, chitosanase Csn174, egg white lysozyme (Roth, Germany), and human chitotriosidase ChT—sourced as described above. Chitosans were dissolved at 1 mg mL$^{-1}$ in 150 mM ammonium acetate buffer pH 4.2 and incubated for 24 h at 37 °C with 30 µg mL$^{-1}$ of enzyme in a final reaction volume of 1 ml, or without enzyme as a control. Concentrated stock solutions of enzymes were used to prevent dilution of the reaction buffer. Degradation products were analysed using UHPLC-ESI-MS$^n$ (Dionex Ultimate 3000RS UHPLC; Thermo Scientific, Milford, USA) via an Acquity UHPLC BEH Amide column (1.7 µM, 2.1 × 150 mm) in combination with a VanGuard precolumn (1.7 µM, 2.1 × 5 mm), both from Waters Corporation (Milford, USA), coupled to an ESI-MS detector (amaZon speed, Bruker Daltonics, Bremen, Germany). Eluent A consisted of 80% (v/v) acetonitrile (Roth, Germany), and eluent B consisted of 20% (v/v) acetonitrile, both supplemented with 10 mM $NH_4HCO_2$ and 0.1% (v/v) formic acid. A column oven temperature of 35 °C was used, and mass spectra were determined in a positive mode over scan range of $m/z$ 50–2000. The parameters for the electrospray ionisation were capillary voltage 4 kV, end plate offset voltage 500 V, nebuliser pressure 1 bar, flow rate of the dry gas 8 L min$^{-1}$, and dry temperature 200 °C. Mass spectra were analysed using Data Analysis 4.1 software (Bruker Daltonics, Bremen, Germany). To investigate the hydrolysis of the polymers, 2 µl of sample was injected to the UHPLC-ESI-MS. The flow rate was adjusted to 0.4 mL min$^{-1}$. Oligomers were separated over a 16 min gradient elution profile: 0-2.5 min isocratic 100% A (80:20 ACN:H2O with 10 mM NH4HCO2 and 0.1% (v/v) HCOOH); 2.5–12.5 min linear from 0% to 75% B (20:80 ACN:H2O with 10 mM NH4HCO2 and 0.1% (v/v) HCOOH); column re-equilibration: 12.5–13.5 min linear from 75% B to 100% A; 13.5-16 min isocratic 100% A.

To assess the transfection efficiency of chitosan/DNA polyplexes, the human breast cancer cell line MCF7 was used (Hölzel Diagnostika GmbH, Germany). Plasmid DNA NTC8685-eGFP (3818 bp) was purchased from Nature Technology Corporation (Lincoln, NE, USA), multiplied in *E. coli* DH5α, and purified using the kit according to the manufacturer's instruction. Purity was confirmed by 1% agarose (AppliChem, Germany) gel electrophoresis, and DNA concentration was measured by Nanodrop (peQlab, Germany). For chitosan/pDNA polyplex preparation, 10 µL of plasmid DNA with a concentration of 0.25 µg µL$^{-1}$ were mixed with 10 µL of chitosan solution previously diluted in 0.1 M of MES buffer (Roth, Germany) to yield the desired $NH_3^+/PO_4^-$ molar charge ratios of 5, 8, 10, or 20. After addition of 35 µL of MES buffer (0.1 M, pH 5.8), samples were vortexed thoroughly and incubated for 30 min at room temperature to allow for self-assembly of the polyplexes. Polyplexes were characterised in terms of size and zeta potential as described above. To assess the stability in the physiological medium used for the transfection assay, 50 µL of freshly prepared polyplexes were added to 1 mL of OptiMEM (pH 6.8), and their

hydrodynamic size was measured at different times of incubation at 37 °C as described above. The binding strength between chitosan and pDNA in the polyplexes was evaluated in a gel retardation assay for which 500 ng of free pDNA and chitosan/pDNA polyplexes in 20 µL TAE buffer supplemented with 3 µL of PicoGreen were loaded onto a 1% agarose gel. The gel was run at 140 V for 45 min; DNA bands were visualised using UV illumination. Gene Ruler™ 1 kb DNA Ladder and O'Gene Ruler™ 1 kb DNA Ladder ready-to-use (Thermo Fisher Scientific) were used as molecular weight markers.

For the in vitro transfection assay, MCF7 cells (100.000 cells/well) were seeded in a 24-well plate using 1 mL/well of RPMI medium (PAA Laboratories, Germany) containing 10% FBS (PAA Laboratories, Germany), 1% penicillin/streptomycin, and 1% L-glutamine (Sigma-Aldrich, Germany). The cells were left to attach overnight at 37 °C and 5% $CO_2$. Polyplexes were prepared 30 min prior to incubation with cells and diluted with serum-free transfection medium (OptiMEM I, Gibco; pH adjusted to 6.8) to a concentration of 2.5 µg pDNA µL$^{-1}$. Lipofectamine 2000 (Invitrogen, Karlsruhe, Germany; 1 µL per 2.5 µg of pDNA) was used as a positive control. Cells were incubated with the polyplexes for 24 h at 37 °C and 5% $CO_2$. Then, the transfection medium was replaced by RPMI medium and incubated for another 24 h for protein expression. Control cells were incubated with medium only. After 48 hours, the transfection efficiency was evaluated qualitatively by the analysis of the GFP fluorescence intensities ($\lambda_{ex} = 488$ nm, $\lambda_{em} = 509$ nm) using a fluorescence microscope (DMi8 automated S/N 409984, Leica Microsystems CMS GmbH, Wetzlar, Germany). Images were recorded using a ×10/0.30 DRY objective and a digital camera (Hamamatsu Flash 4.0-USB3-002560, Japan). Fluorescence intensity was quantified with Tecan ultra Evolution (Safire, Tecan, Salzburg, Austria) in top measurement mode, with manually fixed gain at 100 and ten flashes. Biological experiments were conducted at least in triplicates and with at least three technical replicates per independent experiment. Data were analysed using Tukey multiple comparison tests with a single pooled variance using GraphPad Software Prism v6 (San Diego, CA, USA). Differences were considered statistically significant when $p < 0.05$ (*), $p < 0.01$ (**), or $p < 0.001$ (***).

## Reporting summary

Further information on research design is available in the Nature Portfolio Reporting Summary linked to this article.

## Data availability

Data underlying the Figures are publicly available at Zenodo: https://doi.org/10.5281/zenodo.5163326

## Code availability

Code used to analyse enzymatic mass spectrometric fingerprinting data is publicly available along with the data at Zenodo: https://doi.org/10.5281/zenodo.5163326

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

## Acknowledgements

The authors are grateful to Mengqi Wu and Ayesha Sajid (School of Food Science and Nutrition, University of Leeds, United Kingdom) for the gelation studies, to Dr. Philipp Lemke (Institute for Biology and Biotechnology of Plants, University of Münster, Germany) and professor Dr. Jochen Schmid (Institute for Molecular Microbiology and Biotechnology, University of Münster, Germany) for antimicrobial assays against *Bacillus licheniformis*, and to Franziska Schulze-Bockeloh (Evorion Biotechnologies GmbH, Münster, Germany) for cytotoxicity assays using HUVECs. The research leading to these results has received funding from the European Union's Seventh Framework Programme for research, technological development, and demonstration under grant agreement no. 613931 (B.M.M., F.M.G., A.C.M.) as well as from the German Bundesministerium für Ernährung und Landwirtschaft (BMEL) and its Fachagentur Nachwachsende Rohstoffe (FNR) under project number 22031315 (B.M.M.). We thank Dr. Richard M. Twyman for manuscript editing.

## Author contributions

S.S., J.W., A.N., and T.M. performed the experiments; A.C.M., E.R.M., F.M.G., and B.M.M. supervised the work; all authors contributed to the discussion of results as well as to manuscript writing and editing.

## Funding

## Competing interests

S.S., J.W., A.N., and B.M.M. are inventors of the patent application "Process for the preparation of a non-random chitosan polymer" EP3810660A1 (applicant: University of Münster). The remaining authors declare no competing interests.
