## [Peer Review File · Nature Communications]

Biotechnology-derived chitosans with non-random patterns of acetylation differ from conventional chitosans in their properties and activitiesREVIEWER COMMENTS

Reviewer #1 (Remarks to the Author):

This manuscript is a contribution to the important field of structure/property relationships of polysaccharides relative to sequence. The work is novel and clever, and is a significant contribution. In my view it should be published with minor revisions.

Just two issues of significance for the authors to consider. First, I don't like their "polyglucosamine" nomenclature. Polysaccharides are complex and several beautiful systems of nomenclature have evolved that can describe them completely and concisely. "Polyglucosamine" is not proper polymer nomenclature and does not describe the polysaccharide precisely enough (for example no linkage information) so is not adequate from either perspective. For me the best to use would be the polysaccharide shorthand; $1\rightarrow\beta\text{-D-GlcN}\rightarrow 4$.

Second, it would be a good idea to run one more control experiment; the enzyme catalyzed acylation but without the enzyme. Certainly the amount of acetylation should be small, but it will not be zero, and this chemical reaction will be going on in the background of each enzyme-catalyzed acylation, so it should be quantified.

Other small things:

It really isn't accurate to call chitosan hydrophilic and chitin hydrophobic. Chitin for sure is MORE hydrophobic, but it has two hydroxyls and a carboxamide per six carbons, so definitely would not be classified as hydrophobic. You could contrast them in more sophisticated and accurate ways; chitin is more hydrophobic, chitin is neutral at all physiological pH values whereas chitosan is protonated at ca. pH 6 or below. Terminology like that.

L36 I don't know what you mean by "summarized in the chitosan matrix"

L48 suggest that

L62 Is the sentence starting on this line a hypothesis or is it actually known? Which pharmacokinetics? Citation?

L83 What do you mean by kinetically controlled? By reaction time?

L143 Delete first "the"

L210 In this paragraph you could relate the behavior to different galactomannans in the guar gum family, which also differ in blockiness and therefore in rheological behavior

L256 Delete initial "the"

L316 Another area where it might be interesting to explore structure property relationships for these sequence-modified polymers would be in binding to proteins with which chitosans are known to interact, e.g. by QCMD or SPR

L388 For clarity of reader understanding I suggest you specify which parameters you are talking about here (I presume DP, DS, and PA)

L390 Anhydrosugar units are not truly monomeric in polysaccharides. The monomer for enzyme-cat. polymerization as you know is activated at C-1. Thus "monosaccharide" is a more accurate term.

L390 Don't know what you mean by "a general rule of behavior"

L392 "surprisingly stable" towards what? What do you mean?

L398 methyl esterified (L405 also)

L458 offer game-changing

L481 Paragraph explaining nomenclature needs to be early on in the manuscript

L557 Missing period

L623 Has it ever been discussed in the MS why this pseudomonas bacterium was chosen for the study?

Reviewer #2 (Remarks to the Author):

This paper is a study showing the influence of pattern of acetylation (PA) of chitosan on

different properties. In detail, chitosans with different PA were produced by a noteworthy biochemical enzymatic approach. All the produced chitosans were adequately characterized from the physical-chemical (including NMR, circular dichroism) point of view. Furthermore, it is disclosed the influence of PA on rheological properties, enzymatic degradation, possibility to form nanoparticles and nanocapsules, antibacterial properties and transfection efficiency.

The paper is adequately discussed and supported by a large amount of experimental data. Nevertheless, a couple of additional experiments should be performed to improve the quality of the present paper. Furthermore, a small section of the paper should be rephrased. Additionally, materials and methods section should be completed in some details in order to be easily reproduced.

Details are listed below.

Major comments:

1) Line 153-154 and Figure 2. The dimensions of the block PA-chitosan as function of the chitosan:TPP ratio seem to be stable. However, taking into account the low kcps of all formulations and high PDI (seems close to 0.4) of all formulations, only a limited amount of nanoparticles should be formed by using the block PA-chitosan. I would suggest to report PDI data in a separate figure (within the same panel) in order to appreciate differences between formulations. Moreover, the dimensions vs. intensity (by DLS) of formulations (one formulation fabricated with block PA-chitosan and one fabricated with random-PA chitosan) have to be reported (at least in the supplementary section) and discussed. Furthermore, the influence of the surface charge of nanoparticles as function of NH₂/TPP ratio should be commented.

2) Line 324 and Figure 5. I would suggest to report PDI data in a separate figure (within the same panel) in order to appreciate differences between formulations. Moreover, the dimensions vs. intensity (by DLS) of formulations (one formulation fabricated with block PA-chitosan and one fabricated with random-PA chitosan) have to be reported (at least in the supplementary section) and discussed.

3) Line 326, 329 and Figure 5b. The stability of nanoparticles have to be evaluated after dilution in others physiologically relevant media (e.g. PBS). All nanoparticles after dilution in OptiMEM exhibited a marked increase in PDI. I would suggest to report PDI data in a separate figure (within the same panel) in order to appreciate differences between formulations and different timeframes. Moreover, the dimensions vs. intensity (by DLS) of formulations (one formulation fabricated with block PA-chitosan and one fabricated with random-PA chitosan) have to be reported (at least in the supplementary section) and discussed. Indeed, according to PDI data nanoparticles aggregation and/or precipitation seem to occur. This section of the results should be rephrased.

4) Line 505. Why did you use a sodium acetate buffer at pH 7.5? Other buffers (e.g. phosphate buffer) work more efficiently at that pH.

5) Line 506-508. How did you prepared the mixture with enzymes? Which volumes of each component (enzyme and chitosan solution) did you used?

6) Line 527. Briefly define what D stands for.

7) Line 626. The concentration of chitosan have to be specified. Furthermore, the solvent have to be reported.

8) Line 644-645. The mixing conditions (volumes of each component, i.e. enzyme and chitosan solution) have to be reported.

9) Line 665. The method used with Nanodrop (and the Nanodrop model) have to be reported.

Minor comments:

1) Line 493, 496, 499, 505, 524, 535, 537, 624, 631, 636, 649, 664. All manufacturers of reagents (e.g. propanediol, acetic anhydride, ammonia, sodium acetate, ammonium acetate, HCl, NaNO₂, DCl, D₂O, NYG medium, MTT, PBS, DMSO, acetonitrile, agarose, MES, OptiMEM, PicoGreen, RPMI, FBS) have to be specified.

2) Line 505. The concentration of the buffer have to be specified.

Reviewer #3 (Remarks to the Author):

The present Manuscript by Moerschbacher and co-workers reports the production of biotechnology-derived chitosans with non-random pattern of acetylation (PA) via enzymatic approach and systematically compared with standard chitosans obtained via chemical deacetylation/reacetylation processes. This new set of biopolymers was next investigated in terms of biological properties (antimicrobial activity, cytotoxicity, enzymatic degradation and in vitro transfection). While it is widely recognized that fraction of acetylated units, that is FA, and molecular weight play a cardinal role in modulating chitosan physical-chemical properties as well as final application thereof, little is known about the pattern of two building sugars, i.e. glucosamine and N-acetylglucosamine at variance with, for instance, other marine polysaccharides such as alginates.

Overall results reported in this work are interesting and provide a certain degree of advancement with respect to the actual state of the art on the topic. The Manuscript results well written and reports a nice flow of events with good balance of methodologies employed. The Authors elucidate in a convincing way that the PA is of key importance in tuning chitosan properties. It is opinion of the Referee that the present Manuscript must be enlarged with additional experiments to strengthen the validity of their findings and to build fundamental knowledge about the use of this new set of chitosans. The specific points are the following:

1) Cytotoxicity studies. The Authors used HaCat as model cell line to perform this set of experiments. They must enlarge the panel of cell lines toward at least one human primary cell type. Furthermore, one expects to see in vitro cytotoxicity of chitosan samples at least upon three days of incubation (up to seven days); please extend the time window.

2) Antibacterial activity. The Authors must enlarge this section though anti-biofilm assays using, for instance, *P. aeruginosa* or *S. aureus* as bacterial strain models.

3) While the Authors nicely reported the gelling properties of this new set of chitosans in the presence of triphosphate (TPP) as multivalent anion at the nanoscale, none is reported in relation with macroscopic chitosan gels. Recently it was proved that both FA and molecular weight of chitosans derived from standard chemical re-N-acetylation and gelled by TPP greatly influence mechanical properties of related gels at the macroscale, with potential translation of these biomaterials in tissue engineering or mechanotransduction sectors to mention few examples. It should be of pivotal importance for the Readership to understand how PA would impact on that. Rather than gelation of chitosan in the presence of TPP following standard slow ion diffusion technique, it is opinion of the Referee that the approaches reported by Domard and co-workers to macroscopically gel chitosans showing partial acetylation (exploiting hydroalcoholic media or ammonia solutions; *Biomaterials* 26 (2005) 933–943; *Biomacromolecules* 2005, 6, 653–662) could be more feasible to achieve for Authors from an experimental point of view. Next, standard rheological investigation on macroscopic gels composing of non-random A-type blocks would provide fundamental knowledge on this matter.

Additional minor points:

- 1) Lines 33-34: Please add homogeneous or heterogeneous prior "chemical de-N-acetylation".
- 2) Provide scale bars in Fig. 5c.

point-by-point answers to reviewer comments

Reviewer #1

This manuscript is a contribution to the important field of structure/property relationships of polysaccharides relative to sequence. The work is novel and clever, and is a significant contribution. In my view it should be published with minor revisions.

Just two issues of significance for the authors to consider. First, I don't like their "polyglucosamine" nomenclature. Polysaccharides are complex and several beautiful systems of nomenclature have evolved that can describe them completely and concisely. "Polyglucosamine" is not proper polymer nomenclature and does not describe the polysaccharide precisely enough (for example no linkage information) so is not adequate from either perspective. For me the best to use would be the polysaccharide shorthand; 1)- β -D-GlcpN-(4.

We agree with the reviewer that the term 'polyglucosamine' is unprecise. We also agree with the nomenclature the reviewer suggests. However, in order to improve readability, we have used the precise nomenclature at first mention in the text, where we also define the term polyglucosamine for the remainder of the text (and we slightly rephrased the abstract to avoid the term). We would hope that this compromise between precision and accurateness on the one side and readability and brevity on the other side is acceptable to the reviewer and the editor.

Second, it would be a good idea to run one more control experiment; the enzyme catalyzed acylation but without the enzyme. Certainly the amount of acetylation should be small, but it will not be zero, and this chemical reaction will be going on in the background of each enzyme-catalyzed acylation, so it should be quantified.

We agree with the reviewer that this is an important control experiment. In fact, this was also performed, as shown in Fig. 1a and Supplemental Fig. S1.

Other small things:

It really isn't accurate to call chitosan hydrophilic and chitin hydrophobic. Chitin for sure is MORE hydrophobic, but it has two hydroxyls and a carboxamide per six carbons, so definitely would not be classified as hydrophobic. You could contrast them in more sophisticated and accurate ways; chitin is more hydrophobic, chitin is neutral at all physiological pH values whereas chitosan is protonated at ca. pH 6 or below. Terminology like that.

Again, we agree with the reviewer. We have rephrased sentences in which we refer to the hydrophobicity of GlcNAc (blocks) accordingly.

L36 I don't know what you mean by "summarized in the chitosan matrix"

We have inserted a short explanation.

L48 suggest that

done

L62 Is the sentence starting on this line a hypothesis or is it actually known? Which pharmacokinetics? Citation?

It is a hypothesis or, rather, a conclusion, and we have rephrased the sentence accordingly. Also, we have included examples of aspects of pharmacokinetics which can be expected to be influenced, and we have added a reference.

L83 What do you mean by kinetically controlled? By reaction time?

Yes, we meant to indicate that we can control the reaction by adjusting the reaction conditions such as the incubation time or enzyme concentration. This is now indicated in the manuscript. To avoid misunderstandings, we have deleted the word "kinetically".

L143 Delete first "the"

done (here and elsewhere in the manuscript)

L210 In this paragraph you could relate the behavior to different galactomannans in the guar gum family, which also differ in blockiness and therefore in rheological behavior

Thanks for this suggestion. In fact, we have included this information now in the discussion, with references.

L256 Delete initial "the"

done

L316 Another area where it might be interesting to explore structure property relationships for these sequence-modified polymers would be in binding to proteins with which chitosans are known to interact, e.g. by QCMD or SPR

We again agree with the reviewer. This paper marks the beginning of a new field of chitosan research using novel chitosans, and there are still many aspects that deserve attention, and chitosan-protein interactions are clearly among the most promising ones. We have mentioned this now more explicitly, referring e.g. also to receptor-ligand interactions as another example beyond enzyme-substrate interactions.

L388 For clarity of reader understanding I suggest you specify which parameters you are talking about here (I presume DP, DS, and PA)

done

L390 Anhydrosugar units are not truly monomeric in polysaccharides. The monomer for enzyme-cat. polymerization as you know is activated at C-1. Thus "monosaccharide" is a more accurate term.

done

L390 Don't know what you mean by "a general rule of behavior"

The "general law of behavior" (which we incorrectly cited as "general rule of behavior") of chitosans in aqueous solution has been described by Prof. Alain Domard to explain the role of FA on chitosans' solution properties. We are now more clearly referring to the respective paper.

L392 "surprisingly stable" towards what? What do you mean?

In the interest of brevity, we deleted these two words. (In fact, in this intermediate range between FA 0.25 and 0.5, the physicochemical parameters of chitosans remain more or less constant, as described in the paper of Alain Domard cited above.)

L398 methyl esterified (L405 also)

done here and throughout

L458 offer game-changing

done

L481 Paragraph explaining nomenclature needs to be early on in the manuscript

We have shifted this explanation to the end of the first paragraph of the Results section.

L557 Missing period

done

L623 Has it ever been discussed in the MS why this pseudomonas bacterium was chosen for the study?

This strain was selected because it grows well at slightly acidic pH where the chitosans are soluble and specifically, because the respective assay is well established in our lab. This is now mentioned in the manuscript.

Reviewer #2

This paper is a study showing the influence of pattern of acetylation (PA) of chitosan on different properties. In detail, chitosans with different PA were produced by a noteworthy biochemical enzymatic approach. All the produced chitosans were adequately characterized from the physical-chemical (including NMR, circular dichroism) point of view. Furthermore, it is disclosed the influence of PA on rheological properties, enzymatic degradation, possibility to form nanoparticles and nanocapsules, antibacterial properties and transfection efficiency.

The paper is adequately discussed and supported by a large amount of experimental data. Nevertheless, a couple of additional experiments should be performed to improve the quality of the present paper. Furthermore, a small section of the paper should be rephased. Additionally, materials and methods section should be completed in some details in order to be easily reproduced.

Major comments:

1) Line 153-154 and Figure 2. The dimensions of the block PA-chitosan as function of the chitosan:TPP ratio seem to be stable. However, taking into account the low k_{cps} of all formulations and high PDI (seems close to 0.4) of all formulations, only a limited amount of nanoparticles should be formed by using the block PA-chitosan. I would suggest to report PDI data in a separate figure (within the same panel) in order to appreciate differences between formulations. Moreover, the dimensions vs. intensity (by DLS) of formulations (one formulation fabricated with block PA-chitosan and one fabricated with random-PA chitosan) have to be reported (at least in the supplementary section) and discussed. Furthermore, the influence of the surface charge of nanoparticles as function of NH_2/TPP ratio should be commented.

Yes, the reviewer is right. As mentioned, the number of particles formed by TPP-mediated ionic gelation of block-PA chitosan was lower and the dispersity was higher than when using conventional random-PA chitosans. To more clearly show DPI values, those are now given in a separate graph in the supplement (Fig. S2a). Also, a graph giving the relationship between size and intensity has been added as Supplementary Figure S2b, clearly showing the higher dispersity of the block-PA particles, and this is now also briefly discussed in the main text. The dependence of surface charge (i.e., zeta potential) on NH_2/TPP ratio shown in Figure 2a of the main text is now also commented.

2) Line 324 and Figure 5. I would suggest to report PDI data in a separate figure (within the same panel) in order to appreciate differences between formulations. Moreover, the dimensions vs. intensity (by DLS) of formulations (one formulation fabricated with block PA-chitosan and one fabricated with random-PA chitosan) have to be reported (at least in the supplementary section) and discussed.

done as above (Fig. S6b)

3) Line 326, 329 and Figure 5b. The stability of nanoparticles have to be evaluated after dilution in others physiologically relevant media (e.g. PBS). All nanoparticles after dilution in OptiMEM exhibited a marked increase in PDI. I would suggest to report PDI data in a separate figure (within the same panel) in order to appreciate differences between formulations and different timeframes. Moreover, the dimensions vs. intensity (by DLS) of formulations (one formulation fabricated with block PA-chitosan and one fabricated with random-PA chitosan) have to be reported (at least in the supplementary section) and discussed. Indeed, according to PDI data nanoparticles aggregation and/or precipitation seem to occur. This section of the results should be rephrased.

We agree with the reviewer that in the long run, it will be important to also analyse in more detail the stability of nanoformulations prepared from chitosans with different PA. In fact, we consider assessing the stability of chitosan nanoparticles in the medium used for any bioassay to be an essential control experiment which, however, all too often is neglected in literature. This is why we performed stability assays in the OptiMEM medium used for the transfection studies. However, given that in previous studies (Goycoolea et al. 2016 *Macromol. Biosci.*, 16, 1873; Goycoolea et al 2012 *Colloid Polym Sci* 290:1423–1434), we had found that the colloidal stability of chitosan nanoparticles and nanocapsules strongly depended on F_A and, to a lesser extent, on DP, but NOT on the physiological medium used (RPMI-1640, ECGM, optiMEM), we have not yet extended this study into a systematic comparison of stability in different media. We have explained this now briefly in the manuscript, and we hope that the reviewer and editor will follow this reasoning.

4) Line 505. Why did you use a sodium acetate buffer at pH 7.5? Other buffers (e.g. phosphate buffer) work more efficiently at that pH.

Thank you for spotting this mistake! The word “buffer” has been removed. Of course, an acetate solution at pH 7.5 does not act as a buffer. The sodium acetate is used as a co-substrate, not as a buffering substance.

5) Line 506-508. How did you prepared the mixture with enzymes? Which volumes of each component (enzyme and chitosan solution) did you used?

The concentrations given, for both chitosan and enzymes, are those in the final incubation mixture. Concentrated stocks of enzyme solution were used to prevent dilution of the reaction buffer. This is now indicated in the manuscript.

6) Line 527. Briefly define what D stands for.

D stands for “molecular weight dispersity” as now indicated in the text.

7) Line 626. The concentration of chitosan have to be specified. Furthermore, the solvent have to be reported.

The final concentrations of chitosan used in the assay as well as the preparation of the chitosan stock solution are now given.

8) Line 644-645. The mixing conditions (volumes of each component, i.e. enzyme and chitosan solution) have to be reported.

The final reaction volume is now mentioned in the text.

9) Line 665. The method used with Nanodrop (and the Nanodrop model) have to be reported.

provided as suggested

Minor comments:

1) Line 493, 496, 499, 505, 524, 535, 537, 624, 631, 636, 649, 664. All manufacturers of reagents (e.g. propanediol, acetic anhydride, ammonia, sodium acetate, ammonium acetate, HCl, NaNO₂, DCl, D₂O, NYG medium, MTT, PBS, DMSO, acetonitrile, agarose, MES, OptiMEM, PicoGreen, RPMI, FBS) have to be specified.

provided as suggested

2) Line 505. The concentration of the buffer have to be specified.

The concentration of sodium acetate (3 M) is given. (Note: in the original manuscript, the solution was falsely named as “buffer”.)

Reviewer #3

The present Manuscript by Moerschbacher and co-workers reports the production of biotechnology-derived chitosans with non-random pattern of acetylation (PA) via enzymatic approach and systematically compared with standard chitosans obtained via chemical deacetylation/reacetylation processes. This new set of biopolymers was next investigated in terms of biological properties (antimicrobial activity, cytotoxicity, enzymatic degradation and in vitro transfection). While it is widely recognized that fraction of acetylated units, that is FA, and molecular weight play a cardinal role in modulating chitosan physical-chemical properties as well as final application thereof, little is known about the pattern of two building sugars, i.e. glucosamine and N-acetyl-glucosamine at variance with, for instance, other marine polysaccharides such as alginates.

Overall results reported in this work are interesting and provide a certain degree of advancement with respect to the actual state of the art on the topic. The Manuscript results well written and reports a nice flow of events with good balance of methodologies employed. The Authors elucidate in a convincing way that the PA is of key importance in tuning chitosan properties. It is opinion of the Referee that the present Manuscript must be enlarged with additional experiments to strengthen the validity of their findings and to build fundamental knowledge about the use of this new set of chitosans.

The specific points are the following:

1) Cytotoxicity studies. The Authors used HaCat as model cell line to perform this set of experiments. They must enlarge the panel of cell lines toward at least one human primary cell type. Furthermore, one expects to see in vitro cytotoxicity of chitosan samples at least upon three days of incubation (up to seven days); please extend the time window.

Fulfilling this reasonable suggestion was surprisingly difficult as we had closed down our lab facilities for work with human or animal cells when Dr. Sreekumar, our only expert on such work, had left our group which also meant that we had lost the required expertise. When she finally rejoined our group recently, she thus had to establish a collaboration with colleagues which turned out to be difficult as access to foreign labs was severely restricted in pandemic times. Fortunately, we were finally able to perform, as suggested, a cytotoxicity study using human umbilical vein endothelial cells (HUVECs) as a model for human primary cells (Fig. S5). However unfortunately, we were not able to extend the time of incubation beyond 24 hours. Based on experience in previous studies (e.g. Huang et al. (2004) *Pharm Res* 21: 344-353; <https://doi.org/10.1023/B:PHAM.0000016249.52831.a5>; Loh et al. (2010) *Toxicol Appl Pharmacol* 249: 148-157; <https://doi.org/10.1016/j.taap.2010.08.029>) and on ISO 10993-5 for in vitro cytotoxicity testing of biomaterials (e.g. Jablonska et al. (2021) *Sci Rep* 11: 6628; <https://doi.org/10.1038/s41598-021-85019-6>), we hope, however, that incubation of cells for 24 h, which seems to be the recommended and widely accepted standard, will be acceptable to the reviewer and editor.

2) Antibacterial activity. The Authors must enlarge this section though anti-biofilm assays using, for instance, *P. aeruginosa* or *S. aureus* as bacterial strain models.

For similar reasons as above, we had to perform these addition assays in the lab of colleagues who unfortunately, do not have permission to work with human pathogens so that performing the assay with *P. aeruginosa* or *S. aureus*, as suggested, was not possible. But fortunately, our colleague from the Microbiology Department, Prof. Schmid, is an expert on exopolysaccharides and film-forming bacteria so that, as suggested, we have now been able to add a study on antimicrobial activity against the bio-film forming bacterium *Bacillus licheniformis* (Fig. S4).

3) While the Authors nicely reported the gelling properties of this new set of chitosans in the presence of tripolyphosphate (TPP) as multivalent anion at the nanoscale, none is reported in relation with macroscopic chitosan gels. Recently it was proved that both FA and molecular weight of chitosans derived from standard chemical re-N-acetylation and gelled by TPP greatly influence mechanical properties of related gels at the macroscale, with potential translation of these biomaterials in tissue engineering or mechanotransduction sectors to mention few examples. It should be of pivotal importance for the Readership to understand how PA would impact on that. Rather than gelation of chitosan in the presence of TPP following standard slow ion diffusion technique, it is opinion of the Referee that the approaches reported by Domard and co-workers to macroscopically gel chitosans showing partial acetylation (exploiting hydroalcoholic media or ammonia solutions; *Biomaterials* 26 (2005) 933–943; *Biomacromolecules* 2005, 6, 653-662) could be more feasible to achieve for Authors from an experimental point of view. Next, standard rheological investigation on macroscopic gels composing of non-random A-type blocks would provide fundamental knowledge on this matter.

Extending our gelation studies to the macroscale using a different type of chitosan gel was a real challenge, especially given the restrictions on laboratory work still in force due to the pandemic situation. But we are happy to report that we managed to perform a detailed study on gel formation

in the presence of genipin as a covalent chitosan crosslinker. As predicted by the reviewer, the results are highly interesting and reveal fundamental differences in the kinetics of gel formation as a function of the PA of the chitosan polymer used. Describing this study added about one page of text and one additional figure (Fig. 4) to the manuscript. Given this outcome, we are finally grateful to the reviewer for suggesting this study, and to the editor for insisting on it, even though it has postponed our re-submission of the manuscript considerably. Of course, in future follow-up studies, this can be extended into even broader studies on gelation of the novel chitosans by physical, chemical, and enzymatic methods and on e.g. the mechanical, swelling, and diffusion properties of the resulting gels.

Additional minor points:

1) Lines 33-34: Please add homogeneous or heterogeneous prior “chemical de-N-acetylation”.

done

2) Provide scale bars in Fig. 5c.

provided as suggested

REVIEWERS' COMMENTS

Reviewer #2 (Remarks to the Author):

Authors appropriately addressed all concerns of previous revisions. The study is well conducted and is original. Furthermore, authors significantly improved the overall quality of the present manuscript. Nevertheless, some rheological data of genipin-crosslinked hydrogels should be further discussed.

Specifically, in Figure 4 is reported the peculiar behavior of genipin-crosslinked hydrogels. The addition and discussion of amplitude sweep tests (claimed in Line 733) can provide an additional advancement with respect to the actual state of the art on the topic.

Reviewer #3 (Remarks to the Author):

The manuscript is now ready for publication.

Point-by-Point response.

Reviewer #2 (Remarks to the Author):

Authors appropriately addressed all concerns of previous revisions. The study is well conducted and is original. Furthermore, authors significantly improved the overall quality of the present manuscript. Nevertheless, some rheological data of genipin-crosslinked hydrogels should be further discussed.

Specifically, in Figure 4 is reported the peculiar behavior of genipin-crosslinked hydrogels. The addition and discussion of amplitude sweep tests (claimed in Line 733) can provide an additional advancement with respect to the actual state of the art on the topic.

As suggested, an explanation is added to the manuscript and a relevant figure is added to the supplementary information.